# AlphaFold Database Debiasing for Robust Inverse Folding

**Cheng Tan**[1*]  **Zhenxiao Cao**[2*]  **Zhangyang Gao**[1*]  **Siyuan Li**[3]  **Yufei Huang**[3]  **Stan Z. Li**[3]

[1]Shanghai AI Laboratory  [2]Hong Kong University of Science and Technology
[3]AI Lab, Research Center for Industries of the Future, Westlake University

## Abstract

The AlphaFold Protein Structure Database (AFDB) offers unparalleled structural coverage at near-experimental accuracy, positioning it as a valuable resource for data-driven protein design. However, its direct use in training deep models that are sensitive to fine-grained atomic geometry—such as inverse folding—exposes a critical limitation. Comparative analysis of structural feature distributions reveals that AFDB structures exhibit distinct statistical regularities, reflecting a systematic geometric bias that deviates from the conformational diversity found in experimentally determined structures from the Protein Data Bank (PDB). While AFDB structures are cleaner and more idealized, PDB structures capture the intrinsic variability and physical realism essential for generalization in downstream tasks. To address this discrepancy, we introduce a **De**biasing **S**tructure **A**uto**E**ncoder (DeSAE) that learns to reconstruct native-like conformations from intentionally corrupted backbone geometries. By training the model to recover plausible structural states, DeSAE implicitly captures a more robust and natural structural manifold. At inference, applying DeSAE to AFDB structures produces debiased representations that significantly improve inverse folding performance across multiple benchmarks, and also enhance other structure-conditioned modeling tasks. This work highlights the critical impact of subtle systematic biases in predicted structures and presents a principled framework for debiasing, significantly boosting the performance of structure-based learning tasks like inverse folding.

## 1 Introduction

The advent of highly accurate protein structure prediction, epitomized by AlphaFold2 [1, 2, 3], has fundamentally reshaped the landscape of molecular biology. The resulting AlphaFold Protein Structure Database (AFDB) [4, 5, 6] provides an unprecedented repository of structural information, covering vast swathes of the known proteome with near-experimental resolution. This deluge of data promises to catalyze breakthroughs in data-driven protein analysis and design, offering fertile ground for deep learning models to decipher the complex sequence-structure-function relationship. Yet, despite their accuracy, AlphaFold-predicted structures differ systematically from experimentally determined ones. These differences reflect the inductive biases of the predictive model itself—biases which, although benign for folding, can impair downstream learning tasks.

The challenge posed by this systematic bias becomes particularly salient in applications demanding high structural fidelity, such as inverse folding—the prediction of amino acid sequences compatible with a given protein backbone. This task is exquisitely sensitive to the precise geometric and energetic details of the target structure. To empirically demonstrate this, we investigated the performance of several representative inverse folding models when trained on different structural datasets and on a consistent, held-out set of experimentally determined PDB structures.

---

[*]Equal contribution.

39th Conference on Neural Information Processing Systems (NeurIPS 2025).

Specifically, models such as StructGNN [7], GraphTrans [7], GVP [8], and PiFold [9] were trained independently: once using a curated dataset of high-quality PDB structures, and again using a comparable dataset of high fidelity (pLDDT > 70) AFDB structures. Despite the close structural agreement between the two datasets—as indicated by an average RMSD of approximately 0.2Å(Figure 1a)—the downstream performance on the inverse folding task diverged sharply. As shown in Figure 1(b), models trained on PDB data achieved recovery rates between 34.11% and 43.76%, while those trained on AFDB structures performed markedly worse, with recovery rates ranging from 17.16% to 27.83%. The most extreme degradation was observed for PiFold, which dropped from 43.76% to 17.16% when trained on AFDB data. Intriguingly, we observed a consistent trend: models that performed better on PDB data suffered more acutely when trained on AFDB data—suggesting that stronger models are more prone to overfitting the subtle, non-physical regularities present in AFDB data.

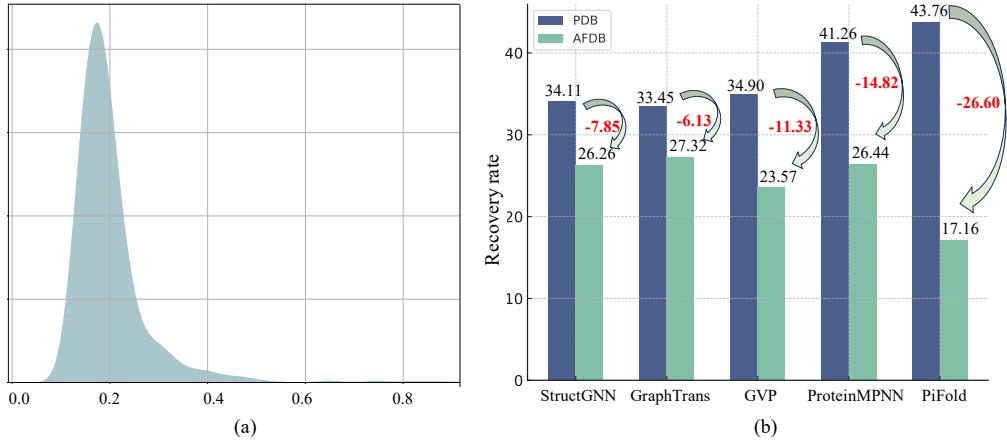

Figure 1: (a) The RMSD distribution of paired structure data from AFDB and PDB. (b) Recovery rate of representative inverse folding methods trained on PDB and AFDB.

These findings provide direct evidence of a distributional shift between AlphaFold-predicted and experimentally observed structures. This performance gap reveals a tangible cost of structural bias: although AFDB structures are highly confident from the model's perspective, they inhabit a displaced region of conformational space—more geometrically idealized, less variable, and ultimately less reflective of the structural diversity sampled in nature. As a result, models trained exclusively on AFDB data fail to generalize reliably to real-world proteins, underscoring the importance of structural realism in downstream design tasks and highlighting the need for principled debiasing strategies.

To intuitively illustrate the divergence between predicted and experimentally determined protein structures, we analyze the distribution of fundamental structural features—such as dihedral angles and inter-atomic distances—across AFDB and PDB. While AFDB structures are generally accurate at the fold level, they tend to occupy a smoother, more regularized region of conformational space—reflecting the inductive biases of the AlphaFold. To bridge this gap and align AFDB structures more closely with experimentally validated conformations, we introduce the Debiasing Structure Autoencoder (DeSAE), a framework trained to reconstruct plausible native conformations from deliberately corrupted backbone geometries. DeSAE learns the structural manifold of experimentally observed proteins, effectively debiasing synthetic structures and improving their downstream utility.

Our main contributions are as follows:

- **Systematic Characterization of Predictive Bias:** We provide, to our knowledge, the first comprehensive identification and quantification of systematic statistical deviations in AFDB when compared against the ensemble of experimental PDB structures.

- **A Debiasing Framework via Manifold Learning:** We introduce DeSAE, a principled framework that learns the manifold of experimentally plausible protein conformations. By training on a denoising objective, DeSAE learns to project AFDB structures onto more realistic structural space.

- **Enhanced Generalization for Inverse Folding:** We rigorously demonstrate that training inverse folding models on DeSAE-debiased AFDB structures leads to consistent and statistically significant improvements in generalization performance across multiple standard benchmarks, validating the practical utility of our debiasing approach.

## 2 Related Work

### 2.1 AlphaFold Database and Its Applications

The development of AlphaFold2 [1] and its subsequent expansion into the AFDB [4] represents a monumental leap in structural biology [10, 11, 12]. AFDB provides open access to millions of high-accuracy predicted protein structures, drastically expanding the known structural proteome far beyond what has been experimentally determined via methods like X-ray crystallography, NMR, or cryo-EM deposited in the PDB [13]. This unprecedented resource has rapidly become foundational for a plethora of applications. Researchers have leveraged AFDB for large-scale functional annotation [14, 15, 16], understanding protein-protein interactions [2, 17], identifying novel drug targets [18, 19], and accelerating structural analysis of complex biological systems [20, 21]. Indeed, structure prediction models like AlphaFold3 [3] have reportedly incorporated AlphaFold2-predicted structures into their training regimens, albeit often with mixed strategies incorporating experimental data.

### 2.2 Structure-based Inverse Folding

Inverse protein folding [22, 23, 24, 25]—the task of predicting an amino acid sequence that will fold into a given three-dimensional backbone structure—is a foundational problem in computational protein design [26, 27, 28, 29, 30]. Historically, this problem was approached using physics-based energy functions [31], but recent advances in deep learning have enabled substantial improvements in both accuracy and scalability [32, 33, 34]. Early methods employed multilayer perceptrons (MLPs) to estimate the probability distribution over 20 amino acids for each residue, based on structural features [35, 36, 37, 38, 39, 40, 41]. Graph-based methods further extend this framework by modeling the protein as a $k$-nearest neighbor graph [42, 43, 44]. StrutGNN and GraphTrans [7] introduced a graph encoder coupled with an autoregressive decoder. GVP [8] leveraged geometric vector perceptrons to jointly learn scalar and geometric vector features. GCA [45] employed global attention mechanisms to capture long-range dependencies. ProteinSolver [46] have addressed partially known sequences, while models like AlphaDesign [47], ProteinMPNN [48], and ESM-IF [49] have achieved strong performance by training on large structural datasets. Several recent works [50, 51] introduce protein language models [52, 53, 54, 55] or surface-based representations [56, 57] to improve inverse folding. Furthermore, many structure-based tasks [58, 59, 60, 61, 62], such as predicting ligand binding sites [63, 64, 65, 66], enzyme commission numbers [67, 68, 69, 70, 71, 72], protein-protein interaction interfaces [73, 74, 75], or post-translational modification [76, 77], also critically depend on precise tertiary structure, often using representations learned by inverse folding models or similar geometric deep learning architectures. Therefore, the fidelity of the structural data used for training these models is paramount. The systematic biases we identify in AFDB could propagate and amplify in such highly sensitive protein structure-based applications, motivating our development of a structural debiasing framework to improve robustness and accuracy of these downstream tasks.

## 3 Preliminaries

A protein can be represented as a sequence of amino acids $S^L = (s_1, s_2, ..., s_L)$ of length $L$, where $s_i \in \mathcal{A}$ and $\mathcal{A}$ denotes the standard amino acid alphabet. The corresponding 3D structure of the protein is defined by the Cartesian coordinates of its backbone atoms—typically including the nitrogen (N), alpha carbon ($C_\alpha$), and carbon (C) atoms for each residue. We denote the backbone conformation as $X^L = \{\boldsymbol{x}_{i,a} \in \mathbb{R}^3 \mid i = 1, \ldots, L; a \in \mathcal{B}\}, \mathcal{B}_i = \{N, C\alpha, C, O\}$.

**Protein Structure Prediction** refers to the task of inferring the 3D structure $X$ from the amino acid sequence $S$. This is typically formulated as learning a function $f : S^L \to X^L$, where the model predicts the atomic coordinates that define the protein's conformation. AlphaFold2 [1], as a notable example, approximates it with remarkable accuracy, making it a cornerstone of structural biology.

**Inverse Folding**, also known as structure-based sequence design, is the complementary problem. Given a target backbone conformation $X$, the goal is to recover a sequence $S$ that is likely to fold into $X$. This task can be expressed probabilistically as modeling the conditional distribution $p(S \mid X)$, or deterministically as learning a function $g : X^L \to S^L$.

As illustrated in Figure 2, inverse folding relies on structure-to-sequence reasoning, in contrast to the sequence-to-structure of prediction models like AlphaFold2. Crucially, this directional shift makes inverse folding models vulnerable to distributional artifacts in the structural data they are trained on.

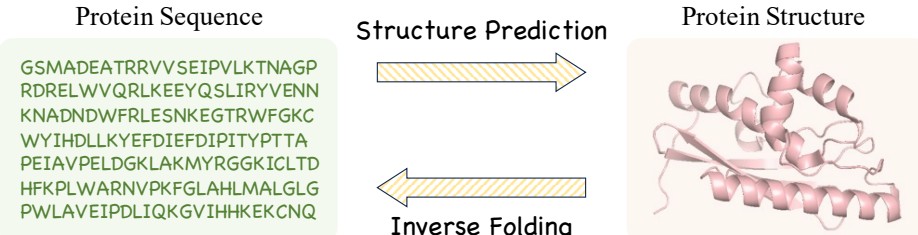

Figure 2: Conceptual comparison between protein structure prediction (sequence-to-structure) and inverse folding (structure-to-sequence).

**Why Inverse Folding Reveals Structural Bias?** We adopt inverse folding as the primary lens through which to study bias in predicted structures. Unlike structure prediction—where small geometric inaccuracies may be tolerable—inverse folding requires a high-fidelity representation of the structural manifold. Any systematic bias in the input structure distribution can distort the learned mapping from structure to sequence. Learning the conditional distribution $p(S|X)$ requires the model to capture fine-grained and context-specific geometric feature.

**The Structural Debiasing Task** The core challenge motivating this work stems from the observation that while models trained on predicted structures $X_{\text{pred}}$ (e.g., from AFDB) may perform well on similar held-out predicted data, their generalization to experimental data $X_{\text{exp}}$ (e.g. from PDB) can be suboptimal. We define the structural debiasing task as learning a transformation $\mathcal{T}$ that debias $X_{\text{pred}}$ into a structure $X'_{\text{pred}} = \mathcal{T}(X_{\text{pred}})$. The primary objective is not merely to alter $X_{\text{pred}}$, but to ensure that downstream models trained or evaluated using $X'_{\text{pred}}$ exhibit improved generalization.

In the specific context of inverse folding, let $\mathcal{F}$ denote an inverse folding model trained on a dataset of structures. When $\mathcal{F}$ is trained on debiased structures $X'\text{pred}$, its effectiveness is evaluated based on its ability to recover amino acid sequences for a held-out set of experimental structures $X\text{exp}$. Let $M(\cdot)$ be a performance metric for the downstream task—e.g., the sequence recovery rate. The transformation $\mathcal{T}$ is considered beneficial if it satisfies the following condition:

$$M(\mathcal{F}(X'_{\text{pred}})) > M(\mathcal{F}(X_{\text{pred}})) \tag{1}$$

with the ideal objective being to approach:

$$M(\mathcal{F}(X'_{\text{pred}})) \to M(\mathcal{F}(X_{\text{exp}})) \tag{2}$$

That is, the model trained on debiased predicted structures should generalize as well as, or nearly as well as, a model trained on high-quality experimental structures.

## 4 Uncovering Systematic Structural Bias in AFDB

### 4.1 Manifestation of Bias: Degraded Performance in Inverse Folding

To empirically evaluate the impact of training exclusively on predicted structures, we constructed a rigorously curated dataset of paired PDB and AFDB entries (see Appendix B.1). We adopt the validation and test splits from CATH 4.2 [78], removing any entries with high sequence similarity with our paired dataset. It is important to note that only the training partition is altered. As previously shown in Figure 1, although the predicted AFDB structures exhibit close agreement with experimental structures, the inverse folding performance of models trained on AFDB structures is markedly worse than those trained on PDB data. This phenomenon is further elucidated by examining the learning dynamics when trained and validated across PDB and AFDB data, as depicted in Figure 3. While models trained on either PDB or AFDB data exhibit superficially similar decreases in training loss (Figures 3a and 3c), a stark divergence emerges in their validation performance. Notably, models trained on AFDB data demonstrate a pronounced difficulty in generalizing, as evidenced by significantly higher or more erratic validation losses on held-out PDB structures (Figure 3d) compared to the robust generalization observed when training and validating on PDB data (Figure 3b).

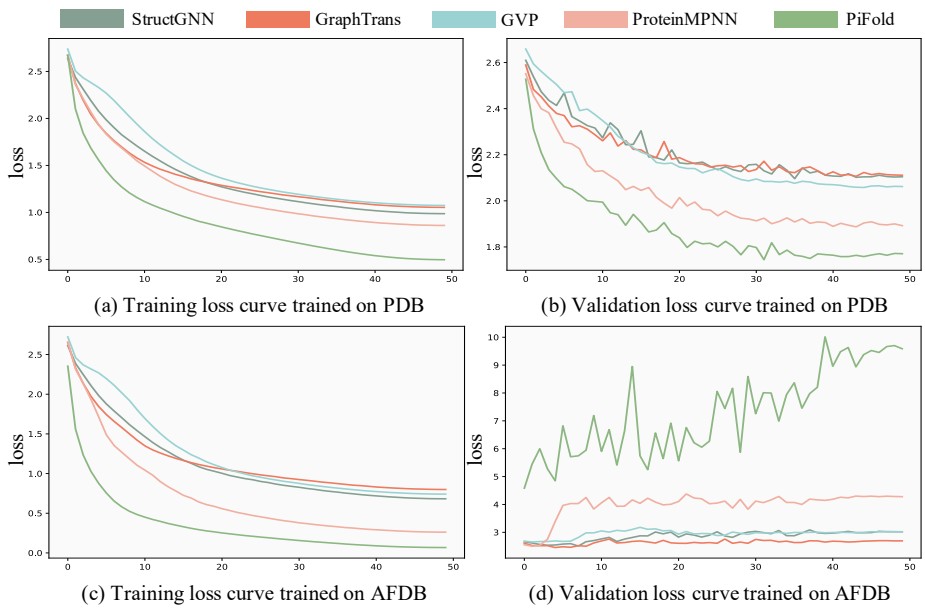

(a) Training loss curve trained on PDB

(b) Validation loss curve trained on PDB

(c) Training loss curve trained on AFDB

(d) Validation loss curve trained on AFDB

Figure 3: Training and validation loss curves for inverse folding models on PDB and AFDB datasets.

## 4.2 Statistical Analysis of Key Structure Features

We performed a comparative statistical analysis of fundamental structural features. We focused on dihedral angles $\phi, \psi, \omega$ and bond lengths (C-C$\alpha$ and N-C$\alpha$) in the main text, as these parameters critically define local protein conformation. More detailed analysis is provided in Appendix E.

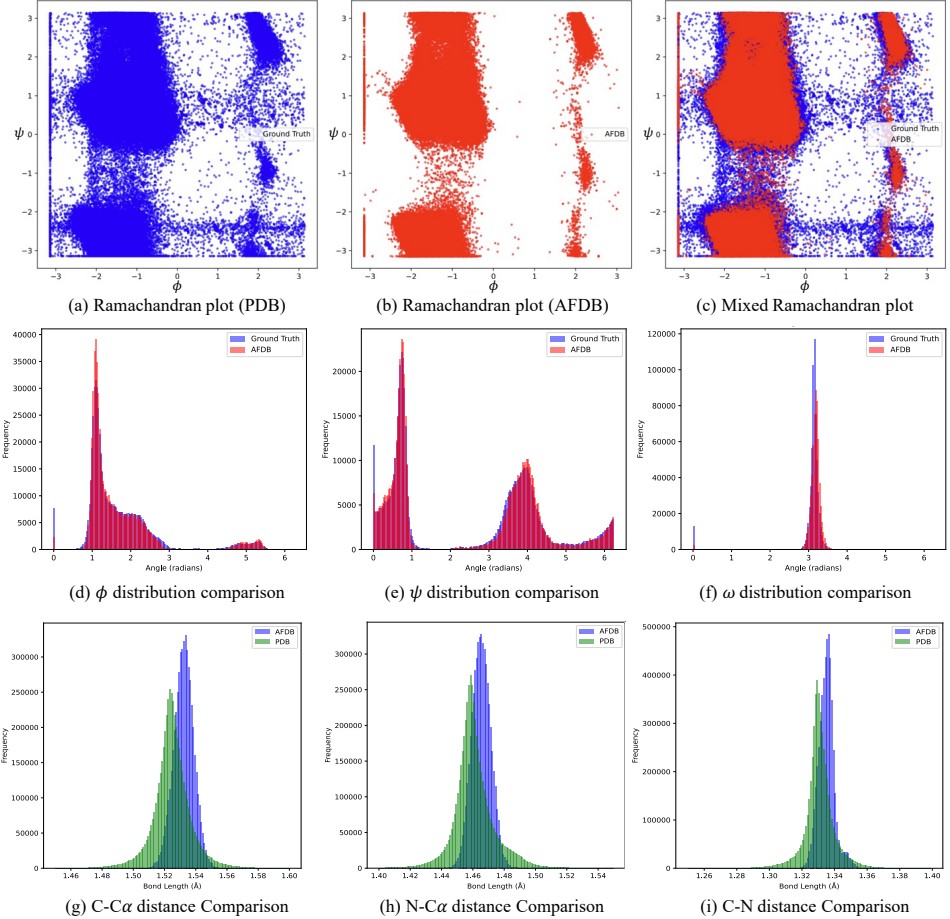

(a) Ramachandran plot (PDB)

(b) Ramachandran plot (AFDB)

(c) Mixed Ramachandran plot

(d) $\phi$ distribution comparison

(e) $\psi$ distribution comparison

(f) $\omega$ distribution comparison

(g) C-C$\alpha$ distance Comparison

(h) N-C$\alpha$ distance Comparison

(i) C-N distance Comparison

Figure 4: Statistical comparison of key structural features between paired PDB and AFDB data.

The Ramachandran plot for PDB (Figure 4a) displays well-defined clusters corresponding to canonical secondary structures ($\alpha$-helices, $\beta$-sheets), but also a significant dispersion of points into less populated and even classically "disallowed" regions. In stark contrast, AFDB structures (Figure 4b) shows considerably tighter, more concentrated clusters within the allowed regions, with a notably sparser population in the disallowed areas. The mixed plot (Figure 4c) vividly demonstrates this: AFDB conformations predominantly occupy the core of the allowed regions, while PDB conformations form a broader envelope and populate peripheral regions more extensively.

Direct comparison of the $\phi, \psi$, and $\omega$ dihedral angle distributions further substantiates these observations. For both $\phi$ and $\psi$ angles, the AFDB distributions exhibit sharper, higher peaks and narrower spreads compared to the PDB distributions. The $\omega$ angle distribution, predominantly centered around $\pi$ radians, also shows a more constrained, needle-like peak for AFDB compared to the slightly broader peak observed for PDB, which can accommodate subtle deviations and cis-peptide bonds. Analysis of backbone bond lengths reinforces this trend of reduced variability in AFDB. The PDB bond length distributions exhibit broader tails, indicative of a greater range of bond distortions reflecting real physical variations or experimental factors.

## 5 Method

The observed "idealization" or regularization bias within AFDB data leads to a narrower distribution of local geometric parameters compared to experimental data. This discrepancy can hinder the development of robust deep learning models for tasks that critically depend on precise atomic details. To address this issue, we propose a simple yet effective **Debiasing Structure Autoencoder (DeSAE)**, illustrated in Figure 5. DeSAE is designed to reconstruct native-like conformations from biased or corrupted structural inputs, thereby learning a generalizable structural manifold.

**Structure Corruption Strategy** We introduce localized perturbations to the backbone coordinates. Specifically, we randomly select a subset of residues and choose one of their backbone atoms. The coordinates of the chosen atom $a^*$ is replaced by the centroid of the remaining three atoms:

$$x'_{i,a^*} = 1/3 \sum_{a \in \mathcal{B}_i \setminus \{a^*\}} x_{i,a}, \tag{3}$$

This strategy forces the model to learn local geometric integrity based on contextual information from the rest of the structure, using the uncorrupted PDB structure as the ground truth for reconstruction.

**DeSAE Architecture** In DeSAE, each residue $i$ is associated with a local frame $T_i(R_i, t_i)$, where $R_i \in \mathrm{SO}(3)$ is a rotation matrix and $t_i \in \mathbb{R}^3$ is a translation vector (typically centered at $C\alpha$). Node features are denoted $h_i \in \mathbb{R}^D$ and edge features between node $s$ and $t$ are $h_{ij} \in \mathbb{R}^D$. In the SE(3) encoder, we employ only the *frame aggregation* layers to capture local geometric interactions. In contrast, the SE(3) decoder utilizes both *frame aggregation* and *frame update* layers, allowing the model to iteratively refine local frames and recover physically consistent backbone geometries.

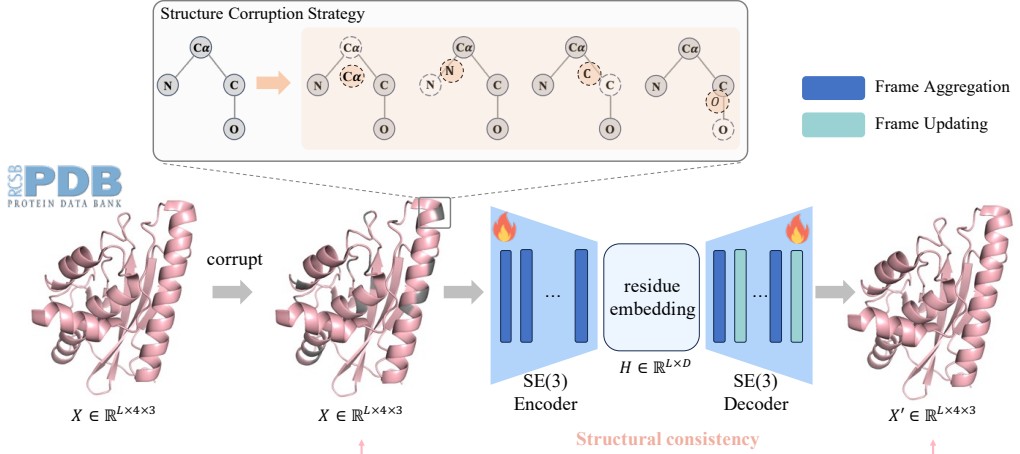

Figure 5: Overview of the DeSAE. The encoder utilizes Frame Aggregation layers to build informative residue and edge representations invariant to global pose. The decoder employs both Frame Aggregation/Updating layers to refine residue frames and reconstruct the debiased protein structure.

**Frame Aggregation**  This module updates node and edge embeddings by processing information in a manner invariant to global rigid transformations. For a pair of nodes $(i, j)$ and their respective frames $T_i, T_j$ at layer $l$, the update process can be summarized as:

$$h_{ij}^{(l+1)}, h_i^{(l+1)} = \text{FrameAgg}(h_{ij}^{(l)}, h_i^{(l)}, T_i, T_j), \tag{4}$$

The aggregation involves several steps:

1. **Projection to latent geometric space**: Node and edge features are projected into a latent 3D space, yielding sets of "virtual atom" coordinates associated with each node and edge:

$$z_i^{(l)} = \mathcal{Z}_\theta(h_i^{(l)}), z_{ij}^{(l)} = \mathcal{Z}_\theta(h_{ij}^{(l)}), \tag{5}$$

   where $\mathcal{Z}_\theta(\cdot) : \mathbb{R}^D \to \mathbb{R}^{m \times 3}$ is a learnable function typically parameterized by MLP followed by reshaping, $m$ represents the number of virtual points.

2. **Augmented edge embedding with relative frame geometry**: To incorporate geometric context, the edge feature is augmented using the relative pose between local frames. The relative transformation is given by $T_{ij} = T_i^{-1} \circ T_j$, and applied to the edge projection:

$$p_{ij} = (T_{ij} \circ z_{ij}^{(l)}) || z_{ij}^{(l)}, \tag{6}$$

3. **Inter-node geometric interaction**: To capture the geometric relationship between the nodes themselves, their latent point clouds $z_i^{(l)}$ and $z_j^{(l)}$ are compared after aligning them within a common reference frame, typically frame $i$. The points $z_j^{(l)}$ are transformed using the relative rotation $R_i^T R_j$ before calculating the geometric dot product with $z_i^{(l)}$.

$$q_{ij} = z_i^{(l)} (R_i^T R_j z_j^{(l)})^T, \tag{7}$$

4. **Edge aggregating**: The final edge embedding $h_{ij}^{(l+1)}$ is aggregated by integrating the augmented geometric features $(p_{ij}^{(l)}, q_{ij}^{(l)})$ and explicit relative frame information:

$$h_{ij}^{(l+1)} = \text{MLP}(p_{ij}, q_{ij}, \text{vec}(R_{ij}), \|t_s - t_t\|), \tag{8}$$

5. **Node aggregating**: The node embedding is aggregated by weighted neighboring nodes:

$$h_i^{(l+1)} = \text{MLP}(h_i^{(l)} + \sum_{j \in \mathcal{N}_i} a_{ij} h_j^{(l+1)}), \tag{9}$$

   where $\mathcal{N}_i$ denotes the set of neighboring nodes, and $a_{ij}$ is the attention weight.

**Frame Updating**  The Frame Updating module, employed exclusively within the decoder, is responsible for refining the residue-specific local frames $T_i^{(l)} = (R_i^{(l)}, t_i^{(l)})$ at layer $l$ to produce updated frames $T_i^{(l+1)} = (R_i^{(l+1)}, t_i^{(l+1)})$ for the subsequent layer.

1. **Rotation updating**: The updated rotation $R_i^{(l)}$ is predicted based on an attention-weighted aggregation of the relative orientations of neighboring frames:

$$\text{vec}(R_i^{(l)}) = \sum_{j \in \mathcal{N}_i} a_{ij}^r \text{vec}(R_{ij}^{(l)}),$$
$$R_i^{(l+1)} = \text{Quat2Rot}(W_r \text{vec}(R_i^{(l)})), \tag{10}$$

   where $a_{ij}^r$ is the learnable attention weight, $W_r \in \mathbb{R}^{4 \times 9}$ projects the vectorized matrix into the 4D space. $\text{Quat2Rot}(\cdot)$ maps a quaternion to its corresponding $3 \times 3$ rotation matrix, detailed in Appendix A.2.

2. **Translation updating**: The updated translation $t_i^{(l+1)}$ is determined by an attention-weighted aggregation of relative positional information derived from neighboring residues:

$$t_i^{(l+1)} = \sum_{j \in \mathcal{N}_i} a_{ij}^t t_{ij}^{(l)}, \tag{11}$$

   where $a_{ij}^t$ is the learnable attention weight. The coordinates are determined by $x_i = T_i^{(l)} \circ h_i$.

**Structural consistency**  We employ the structure loss inspired by Chroma [79], which aims to directly minimize the deviation between the predicted and reference backbone conformations. The detailed loss function is provided in the Appendix A.1.

## 6  Experiments

We begin by curating a paired dataset of experimentally determined structures from PDB and their corresponding AFDB data (see Appendix B). We first pretrain DeSAE with the proposed structure corruption strategy that corrupt randomly 10% residues. The trained DeSAE was then applied to the AFDB structures to produce a "Debiased AFDB" dataset. **Due to its simple design, DeSAE has only 5.9M parameters and is capable of processing process about 20k AFDB structures in 3 minutes on a single NVIDIA A100 GPU.** To assess the impact of this debiasing, we evaluated these datasets on a downstream inverse folding task. Specifically, the inverse folding model was trained independently using three distinct datasets: (1) PDB, (2) AFDB, and (3) Debiased AFDB. Further specifics on model architectures, and experimental parameters are provided in Appendix C.

### 6.1  Does Debiasing Work?

We train models on three datasets and evaluate them using the validation and test sets of CATH 4.2 [78]. A critical aspect of structural debiasing is the preservation of the overall protein fold. Figure 6(a) presents the RMSD distribution remain concentrated at low values, and even seems unchanged compared to Figure 1(a), indicating that the debiasing process effectively debias AFDB without introducing significant global distortions. More importantly, the Debiased AFDB enables substantial improvements in sequence recovery across five inverse folding models (Figure 6(b)).

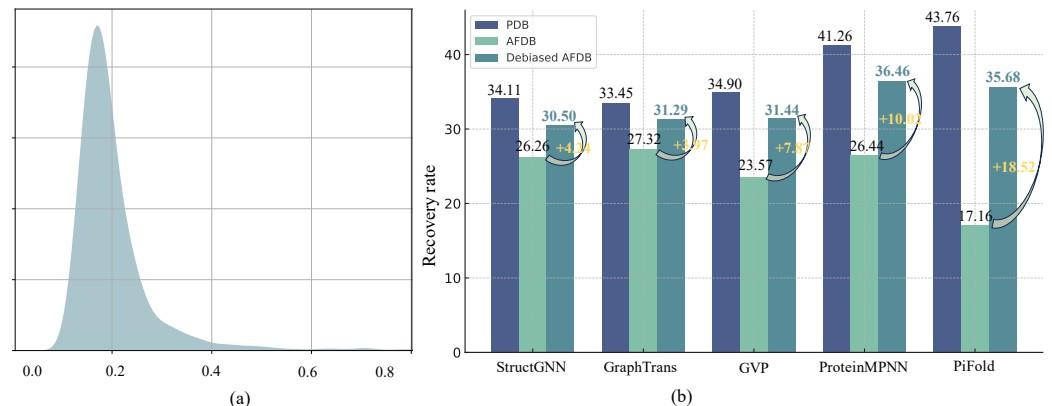

Figure 6: (a) The RMSD distribution of paired structure data from Debiased AFDB and PDB. (b) Recovery rate of inverse folding methods trained on three datasets and evaluated on CATH 4.2.

Training dynamics on the Debiased AFDB are more favorable than on the original AFDB. As depicted in Figure 7, the validation loss exhibits a decay thrend over training epochs. While it remains less stable than training on PDB, it is significantly more stable and lower in magnitude than that observed for AFDB-trained models (see Figure 3). These results collectively affirm that our DSAE not only preserves structural accuracy but also enhances the learnability of inverse folding models.

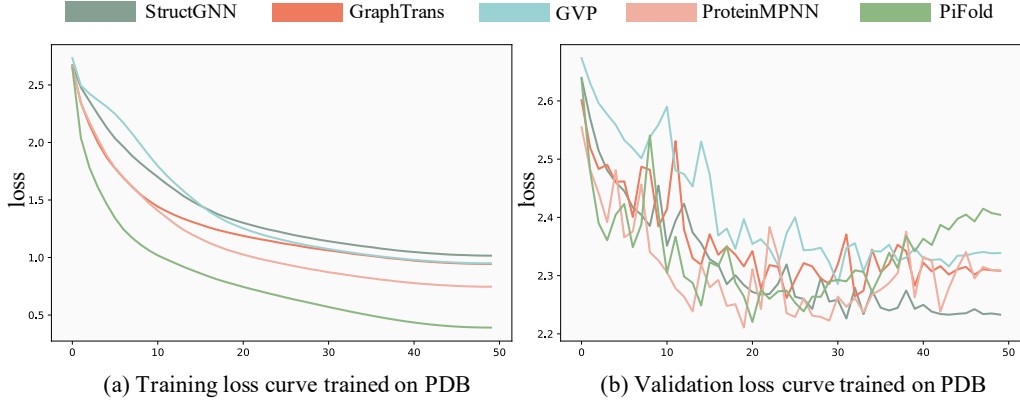

(a) Training loss curve trained on PDB  (b) Validation loss curve trained on PDB

Figure 7: Training and validation loss curves for inverse folding models on Debiased AFDB.

## 6.2 Does Debiasing Generalizable?

To assess the generalization capability of our Debiased AFDB, we use the trained model further evaluate on TS50 and TS500 test sets, which represent diverse and difficult inverse folding challenges. As shown in Figure 8, the performance advantages of Debiased AFDB persist on both TS50 and TS500 benchmarks. Similarly, results on the CATH 4.3 test set reproted Table 1 reinforce this trend. Across all five inverse folding models training on Debiased AFDB leads to consistent improvements in sequence recovery compared to training on original AFDB structures. These results strongly indicate that our debiasing methodology effectively enhances the structural realism of AFDB data, enabling models to learn more generalizable sequence-structure relationships.

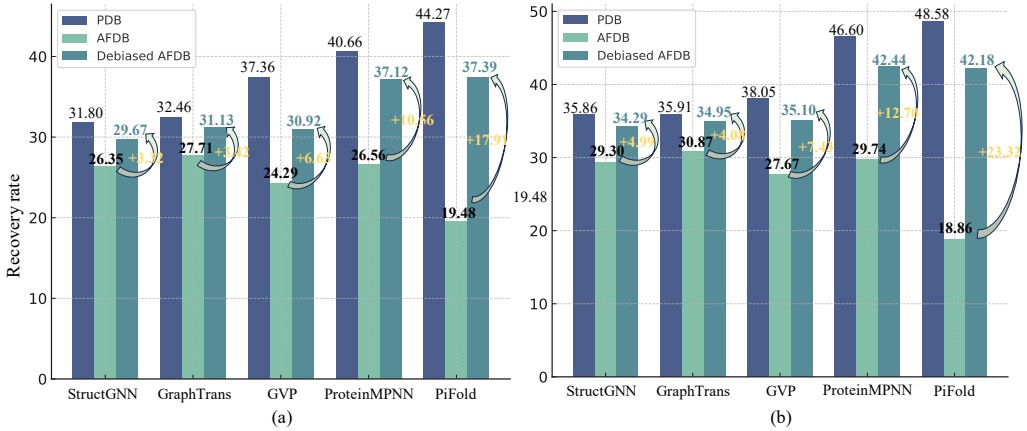

Figure 8: Recovery rate of inverse folding methods trained on PDB, AFDB, and Debiased AFDB. (a) Test on TS50 benchmark. (b) Test on TS500 benchmark.

## 6.3 Does Larger Scale Pretraining DeSAE Benefit Downstream Task?

The default DeSAE model is pretrained on the PDB subset of our paired dataset, which comprises approximately 20,000 structures. To explore the benefits of larger-scale pretraining, we further pretrain DeSAE on the full PDB dataset and apply our debiasing pipeline to the AFDB data of the paired dataset. We refer to this extended dataset as Debiased AFDB-XL. As shown in Table 1, Debiased AFDB-XL yields modest improvements over the Debiased AFDB. However, the overall gains are limited. A possible reason is that DeSAE primarily captures local geometric features, and thus may not fully benefit from additional global or large-scale structural diversity in pretraining.

Table 1: Recovery rate (%) of inverse folding methods trained on different data and test on the CATH 4.3 test set. The footnote colored in blue indicates the improvement of Debiased AFDB over the AFDB, colored in green indicates the improvement of Debiased AFDB-XL over the Debiased AFDB.

| Method | PDB | AFDB | Debiased AFDB | Debiased AFDB-XL |
|--------|-----|------|---------------|------------------|
| StructGNN | 32.62 | 26.04 | 29.99$_{(+3.95)}$ | 30.57$_{(+0.58)}$ |
| GraphTrans | 32.84 | 27.05 | 30.62$_{(+3.57)}$ | 30.96$_{(+0.34)}$ |
| GVP | 34.81 | 24.35 | 31.15$_{(+6.80)}$ | 31.66$_{(+0.51)}$ |
| ProteinMPNN | 42.33 | 26.48 | 35.22$_{(+8.74)}$ | 35.55$_{(+0.33)}$ |
| PiFold | 43.74 | 17.74 | 35.38$_{(+17.64)}$ | 35.76$_{(+0.38)}$ |

More detailed experimental analysis and statistics are provided in Appendix D.

## 7 Conclusion and Limitation

In this work, we identified and addressed a critical challenge in leveraging the vast AFDB for training deep learning models sensitive to precise atomic details, particularly for the task of inverse folding. We demonstrated that a systematic bias exists within AFDB structures. To mitigate this, we propose DeSAE to reconstruct native-like experimental structures from corrupted inputs. Our extensive experiments consistently showed that our debiasing pipeline effectively debiases AFDB without introducing significant global distortions. One limitation of our approach is its focus on backbone geometries, potentially overlooking higher-order structural details such as sidechain orientations.

## Acknowledgement

This work was supported by 2025 Local Supporting Project (PJ-PRJ24DATA001) by Shanghai Artificial Intelligence Laboratory, National Science and Technology Major Project (No. 2022ZD0115101), National Natural Science Foundation of China Project (No. 624B2115, No. U21A20427), Project (No. WU2022A009) from the Center of Synthetic Biology and Integrated Bioengineering of Westlake University, Project (No. WU2023C019) from the Westlake University Industries of the Future Research Funding.

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

# A  Technical Details

## A.1  Structure Consistency Loss

Inspired by Chroma [79], we supervise our model with five complementary loss terms:

$$\mathcal{L} = \mathcal{L}_{global} + \mathcal{L}_{fragment} + \mathcal{L}_{pair} + \mathcal{L}_{neighbor} + \mathcal{L}_{distance} \tag{12}$$

To define these terms, let $X \in \mathbb{R}^{n \times 3}$ be the ground-truth backbone coordinates of $n$ residues and $\hat{X} \in \mathbb{R}^{n \times 3}$ their predictions. We first introduce the basic aligned RMSD loss:

$$\mathcal{L}_{aligned}(\hat{X}, X) = \|\text{Align}(\hat{X}, X) - X\| \tag{13}$$

where $\text{Align}(\hat{X}, X)$ rigidly aligns the prediction to the target before measuring deviation. Thus, the five loss terms are defined based on the aligned RMSD loss as follows:

- **Global Loss ($\mathcal{L}_{global}$):** Apply $\mathcal{L}_{aligned}$ to the full backbone of each residue, treating $X$ as $[n, 4, 3]$ (four atoms per residue).
- **Fragment Loss ($\mathcal{L}_{fragment}$):** For each residue, consider its sequential fragments of $c = 7$ residues centered around each residue; $X$ has shape $[n, c, 4, 3]$.
- **Pair Loss ($\mathcal{L}_{pair}$):** For each of the $K = 30$ nearest-neighbor residue pairs, measure alignment error over $c$ fragments on both residues, i.e. shape $[n, K, c \cdot 2, 4, 3]$.
- **Neighbor Loss ($\mathcal{L}_{neighbor}$):** Compute RMSD over the four backbone atoms of the $K$ nearest neighbors for each residue, with $X$ shaped $[n, K, 4, 3]$.

Finally, the **Distance Loss ($\mathcal{L}_{distance}$)** enforces correct inter-residue distances using an MSE objective:

$$\mathcal{L}_{distance} = \|\text{Dist}(\hat{X}) - \text{Dist}(X)\| \tag{14}$$

where $\text{Dist}(X) \in R^{n \times n}$ is the matrix of pairwise $C_\alpha$-$C_\alpha$ distances derived from the coordinates $X$.

We compute each of these five losses at every decoder layer and average them over the $L$ layers. Empirically, this multi-scale supervision is crucial to achieve good global structural fidelity.

## A.2  Quat2Rot Function

Let $q = (w, x, y, z)$ be the quaternion, then:

$$R(q) = \begin{pmatrix} w^2 + x^2 - y^2 - z^2 & 2(xy - wz) & 2(xz + wy) \\ 2(xy + wz) & w^2 - x^2 + y^2 - z^2 & 2(yz - wx) \\ 2(xz - wy) & 2(yz + wx) & w^2 - x^2 - y^2 + z^2 \end{pmatrix}, \tag{15}$$

Equivalently, each component of the rotation matrix is given by:

$$
\begin{aligned}
R_{00} &= w^2 + x^2 - y^2 - z^2, \\
R_{01} &= 2(x\,y - w\,z), \\
R_{02} &= 2(x\,z + w\,y), \\
\\
R_{10} &= 2(x\,y + w\,z), \\
R_{11} &= w^2 - x^2 + y^2 - z^2, \\
R_{12} &= 2(y\,z - w\,x), \\
\\
R_{20} &= 2(x\,z - w\,y), \\
R_{21} &= 2(y\,z + w\,x), \\
R_{22} &= w^2 - x^2 - y^2 + z^2.
\end{aligned}
\tag{16}
$$

# B    Dataset Details

This section details the curation of datasets used for training the DeSAE and for evaluating its efficacy in downstream tasks, particularly inverse folding. Rigorous procedures were implemented to ensure data quality and prevent information leakage between training and test sets.

## B.1    DeSAE Training Dataset: Paired AFDB-PDB Structures

The DeSAE model is trained to learn a mapping from potentially biased predicted structures to conformations more representative of experimental observations. To facilitate this, we constructed a dataset of paired protein structures, where each pair consists of:

1. An AlphaFold-predicted structure sourced from the AFDB [4, 1].
2. Its corresponding experimentally determined structure from the PDB [13].

The curation process for this paired dataset involved several steps:

1. Initial Pairing: We identified all PDB entries that have a corresponding prediction available in the AFDB based on UniProt accession numbers.
2. Quality and Consistency Filtering: To ensure a meaningful structural correspondence and high-quality predictions, we applied the following filters:
    - The AlphaFold prediction must exhibit a mean predicted Local Distance Difference Test (pLDDT) score greater than 70. The choice of a pLDDT score > 70 is a widely adopted convention in the field for filtering high-confidence predictions from AlphaFold [14]. By setting the threshold at >70, we ensure that the AFDB structures used in our study are of high quality while maximizing dataset coverage across the proteome.
    - The sequence lengths of the AFDB-predicted structure and the PDB experimental structure must be identical.
3. Residue-Level Matching: By enforcing identical sequence lengths and originating from the same protein, we ensure a direct residue-to-residue mapping between the predicted and experimental structures within each pair.

This curation process yielded a high-quality dataset of **19,392 AFDB-PDB paired structures**.

## B.2    Downstream Task Evaluation: Inverse Folding Datasets

To evaluate the impact of DeSAE-debiasing on inverse folding performance, we prepared three distinct structural datasets derived from our curated pairs:

1. PDB Dataset: The experimental structures from the 19,392 PDB entries.
2. AFDB Dataset: The corresponding AlphaFold predictions from the AFDB.
3. Debiased AFDB Dataset: The AFDB structures after being processed by trained DeSAE model.

Inverse folding models are trained and evaluated separately on these three structural datasets to quantify the effect of debiasing.

## B.3    Benchmark Test Sets and Data Leakage Prevention

To assess the generalization capabilities of inverse folding models trained on the aforementioned datasets, we utilized several established benchmark test sets: CATH 4.2 [78], TS50, TS500, and CATH 4.3 [78]. To ensure that our benchmark evaluations are not compromised by data leakage from the DeSAE training set or the inverse folding training sets, we implemented a strict sequence similarity filtering protocol. Using MMseqs2 [80, 81], we removed any protein structure from these benchmark test sets if its sequence exhibited more than 90% sequence identity to any sequence present in our comprehensive 19,392-structure paired dataset used for DeSAE training. Following this filtering, our final benchmark test sets comprise: 893 structures from CATH 4.2, 38 from TS50, 382 from TS500, and 1575 from CATH 4.3. Performance on these carefully curated, non-overlapping test sets provides a robust measure of model generalization.

# C  Experiment Details

Our experimental methodology involves a two-stage process, followed by downstream task evaluation, as illustrated in Figure 9. First, DeSAE is trained using experimental PDB structures. Second, the trained DeSAE is employed to process and debias AFDB structures. Finally, inverse folding models are trained on the original PDB, original AFDB, and the DeSAE-debiased AFDB structures to evaluate the impact of our debiasing approach.

**Stage 1: Pretraining DeSAE on PDB.**    To initialize our debiasing autoencoder, we trained DeSAE exclusively on the PDB portion of our paired dataset. Training spanned 60 epochs with an initial learning rate of $1 \times 10^{-3}$, a batch size of 16, and a CosineAnnealingLR scheduler to anneal the learning rate. The SE(3) encoder comprised eight equivariant layers, and the decoder comprised six layers, each with a hidden dimensionality of 128. In each epoch, we randomly corrupted 10% of the residues in every structure to enforce robustness against atomic perturbations.

**Stage 2: Generating a Debiased AFDB.**    After pretraining, we applied the learned DeSAE as a preprocessing step to our AFDB structures. Specifically, we passed the AFDB entries through DeSAE to mitigate systematic biases in the predicted coordinates, producing a "Debiased AFDB" dataset. Together with the original AFDB and PDB sets, this yielded three distinct paired training datasets for downstream evaluation: (1) PDB, (2) AFDB, and (3) Debiased AFDB.

**Stage 3: Inverse Folding Evaluation.**    To assess the impact of debiasing on inverse folding performance, we trained separate inverse-folding models on each of the three datasets. Each model was trained for 50 epochs with a learning rate of $1 \times 10^{-3}$ and a batch size of 32. The performance of these three differently trained inverse folding models was then compared on the independent benchmark test sets detailed in Appendix B.

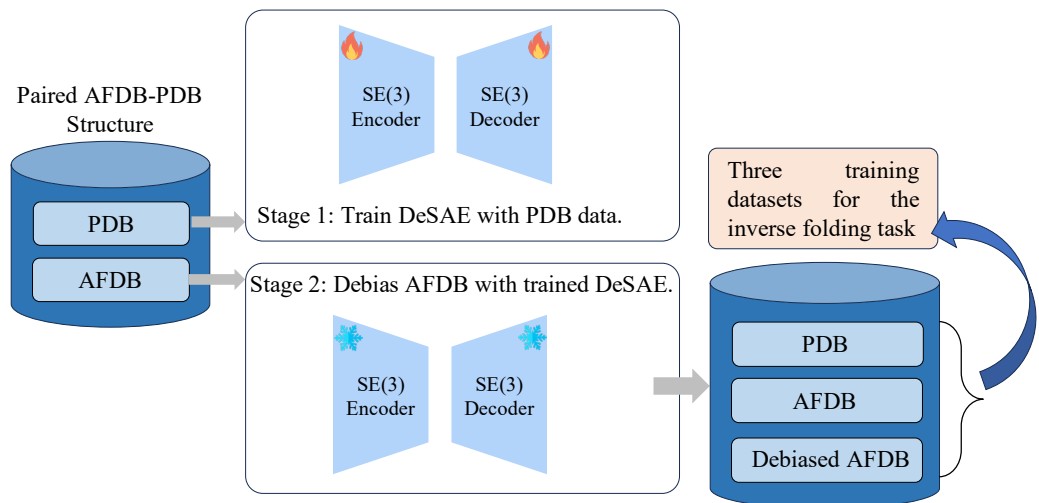

Figure 9: Overview of our experimental pipeline. Stage 1 pretrains DeSAE on PDB data; Stage 2 applies DeSAE to AFDB to obtain Debiased AFDB; Stage 3 trains and evaluates inverse-folding models on PDB, AFDB, and Debiased AFDB datasets for comparison.

**Baseline**    To rigorously evaluate our approach, we compare against five state-of-the-art inverse-folding methods: StructGNN [7], GraphTrans [7], GVP [8], ProteinMPNN [48] and PiFold [9]. Each baseline is trained under identical conditions on three distinct datasets: (1) the original AFDB structures, (2) experimentally determined PDB structures, and (3) our debiased AFDB set.

As shown in the main text, models with higher expressive capacity—such as PiFold perform better on PDB dataset, but are dramatically underperformed on AFDB dataset. These models are indeed designed to capture subtle geometric patterns and fine-grained correlations within structural data. When trained on high-fidelity experimental structures from the PDB, these models benefit from the

natural variability and physical realism inherent in the data, leading to strong generalization.

However, AFDB structures, while accurate at the fold level, exhibit systematic geometric regularities due to AlphaFold's inductive biases—such as smoother dihedral angle distributions and reduced local structural noise. High-capacity models can inadvertently latch onto these superficial statistical cues, treating them as meaningful signals. This leads to distribution-specific overfitting: the model internalizes patterns that are predictive only within the synthetic AFDB manifold, rather than learning robust, physically grounded sequence-structure mappings. In contrast, lower-capacity models may act as a natural form of regularization, failing to fully absorb these subtle biases and thus generalizing marginally better.

**Metric**    We primarily employed the sequence recovery rate that measures the residue-wise accuracy:

$$\text{Recovery Rate} = \frac{1}{L} \sum_{i=1}^{L} \mathbb{I}(s_i = s_i'), \tag{17}$$

where $S_{pred} = (s_1', s_2', ..., s_L')$ is the output sequence of inverse folding methods. In addition to recovery rate, we also considered perplexity as a complementary metric. Perplexity is widely used in sequence modeling tasks:

$$\text{Perplexity} = \exp\left(-\frac{1}{L} \sum_{i=1}^{L} \log P(s_i|X)\right) \tag{18}$$

## D    Additional Experiments

### D.1    Evaluating Generalization to AlphaFold Structures

To further investigate the distinct characteristics learned by models trained on different structural data sources, we conducted an auxiliary experiment. In this scenario, we evaluated the performance of inverse folding models trained on PDB, AFDB, and Debiased AFDB structures, **but this time, the test set was constructed using AlphaFold-predicted structures** corresponding to the CATH4.2 benchmark (termed "AFDB-version CATH4.2 test set"). This setup allows us to assess how well models generalize to the specific structural distribution of AlphaFold predictions themselves.

The results, presented in Table 2, reveal several interesting trends. Unsurprisingly, models trained directly on raw AFDB data consistently achieve the highest recovery rates on this AFDB-version test set. This is expected, as the training and testing distributions are perfectly matched, and the models have effectively learned the idiosyncratic features and potential biases inherent in AlphaFold's predictions. Interestingly, models trained on Debiased AFDB structures achieve the second-best performance. While the debiasing process aims to shift AFDB structures towards the PDB distribution, these structures evidently retain sufficient AFDB-like characteristics to perform well when tested on AFDB predictions, outperforming models trained solely on PDB data for most architectures.

Notably, models trained exclusively on experimental PDB structures, despite having no exposure to AlphaFold's specific predictive patterns or biases during training, still demonstrate commendable generalization to the AFDB-version test set. For instance, PiFold trained on PDB achieves a recovery rate of 44.31%. This suggests that experimental structures encapsulate fundamental sequence-structure relationships that possess inherent generalizability, even to predicted structures that might deviate subtly from experimental reality. This observation reinforces the concept that PDB structures, while potentially "noisier," represent a more foundational and broadly applicable structural truth. The strong performance of AFDB-trained models on AFDB test data further underscores the existence of a distinct "AFDB manifold" which models can readily learn, but which may not perfectly align with the manifold of experimental structures.

### D.2    Differential Impact of Large-Scale DeSAE Pretraining on Recovery and Perplexity

Our previous analysis in Table 1 indicated that pretraining DeSAE on an expanded PDB dataset provided Debiased AFDB-XL which yielded only marginal improvements in the sequence recovery

Table 2: Recovery rate (%) of inverse folding methods trained different data and test on the AFDB-version CATH4.2 test set.

| Method | PDB | AFDB | Debiased AFDB |
|--------|-----|------|---------------|
| StructGNN | 34.21 | 37.83 | 36.16 |
| GraphTrans | 33.74 | 43.45 | 36.64 |
| GVP | 35.60 | 45.22 | 37.63 |
| ProteinMPNN | 41.87 | 76.83 | 45.34 |
| PiFold | 44.31 | 62.87 | 46.35 |

rate. To gain a more nuanced understanding of the effects of this larger-scale pretraining, we further evaluated the models using the perplexity metric, where lower values signify better performance by indicating higher confidence in the predicted amino acid probabilities.

The results, presented in Table 3, reveal a more pronounced benefit of using Debiased AFDB-XL when assessed by perplexity. Across all inverse folding architectures tested on the CATH 4.3 set, models trained on Debiased AFDB-XL consistently achieved lower (i.e., better) perplexity scores compared to those trained on the standard Debiased AFDB. This divergence suggests that while the additional structural information learned by DSAE-XL from the larger PDB corpus may not substantially alter the single best amino acid prediction at each position (thus leading to minor changes in recovery rate), it does refine the overall probability distribution over possible amino acids. The model becomes more confident in its correct predictions and less confident in incorrect ones, leading to lower perplexity.

Table 3: Perpelxity metric of inverse folding methods trained on different data and test on the CATH 4.3 test set.

| Method | PDB | AFDB | Debiased AFDB | Debiased AFDB-XL |
|--------|-----|------|---------------|------------------|
| StructGNN | 8.81 | 11.90 | 10.64 | 9.67 |
| GraphTrans | 8.75 | 14.09 | 10.55 | 9.56 |
| GVP | 8.07 | 16.59 | 10.14 | 9.25 |
| ProteinMPNN | 7.07 | 11.70 | 10.19 | 8.77 |
| PiFold | 6.42 | 13.83 | 10.39 | 8.43 |

### D.3 Experimental sensitivity to the mask ratio

In the training stage of DeSAE, we specified the mask ratio to 0.1. However, we did not elaborate here on the rationale behind selecting this parameter or on the sensitivity of downstream tasks to it. To empirically validate the robustness of our approach, we conducted an ablation study to assess how performance changes with different corruption rates. We trained DeSAE using rates of 10%, 20%, and 30% and then evaluated the performance of downstream inverse folding models trained on the resulting debiased datasets.

Table 4: Performance under different mask ratios. Values in parentheses indicate change relative to mask=10%.

| Model | Mask=10% | Mask=20% | Mask=30% |
|-------|----------|----------|----------|
| StructGNN | 30.50 | 30.45 (-0.05) | 32.06 (+1.56) |
| GraphTrans | 31.29 | 32.53 (+1.24) | 32.68 (+1.39) |
| GVP | 31.44 | 34.27 (+2.83) | 34.14 (+2.70) |
| ProteinMPNN | 36.46 | 38.64 (+2.18) | 38.55 (+2.09) |
| PiFold | 35.68 | 35.66 (-0.02) | 38.41 (+2.73) |

As result shown in 4, the results clearly demonstrate that the performance is not critically sensitive to the 10% rate. For most models, performance is stable or even improves with higher corruption rates. This indicates that our method is robust and that forcing the model to solve a more challenging

reconstruction task may lead it to learn an even more generalizable representation of the structural manifold. The 10% rate used in the paper is a conservative but effective choice, and the method's strong performance across a range of rates confirms its stability.

# E   Structure Features Analysis

## E.1   Quantitive Analysis

**Bond length**   To quantitatively substantiate our hypothesis regarding systematic differences between AFDB and PDB structural ensembles, we performed a detailed statistical analysis of canonical backbone bond lengths: C$\alpha$-N, C-C$\alpha$, O-C, and N-C. The comparative statistics are summarized in Table 5. While the mean values for these bond lengths are highly comparable between AFDB and PDB structures, their variances differ substantially. Across all four analyzed bond types, PDB structures consistently exhibit significantly larger variances.

Table 5: Comparison of bond length statistics between AFDB and PDB structures.

|  | C$\alpha$-N | C-C$\alpha$ | O-C | N-C |
|---|---|---|---|---|
| AFDB | $1.4654\pm3.68\times10^{-5}$ | $1.5323\pm4.11\times10^{-5}$ | $1.2305\pm1.40\times10^{-5}$ | $1.3351\pm2.39\times10^{-5}$ |
| PDB | $1.4610\pm\mathbf{16.95}\times10^{-5}$ | $1.5248\pm\mathbf{15.48}\times10^{-5}$ | $1.2336\pm\mathbf{14.14}\times10^{-5}$ | $1.3308\pm\mathbf{9.57}\times10^{-5}$ |

This pattern of significantly tighter bond length distributions in AFDB structures suggests a higher degree of geometric regularity and uniformity compared to experimental structures. It implies that AFDB predictions may not fully capture the natural conformational fluctuations and slight deviations present in the PDB, potentially reflecting an "over-regularization" or a confinement to a narrower, more idealized region of the conformational landscape. This quantitative finding lends strong support to our premise that AFDB structures possess distinct statistical properties that can contribute to the observed generalization gap in downstream tasks sensitive to such fine-grained structural details.

**Angle distribution**   To quantitatively assess the geometric differences between predicted and experimental structures, we analyzed the distributions of key backbone angles within our paired dataset, comparing structures from AFDB against their corresponding PDB entries. We focused on the distributions of three backbone dihedral angles $(\phi,\psi,\omega)$ and three backbone bond angles $(\alpha,\beta,\gamma)$ defined as follows for residue $i$:

Dihedral angles:

- $\phi_i$, angle defined by atoms $C_{i-1}$-$N_i$-$C\alpha_i$-$C_i$.
- $\psi_i$, angle defined by atoms $N_i$-$C\alpha_i$-$C_i$-$N_{i+1}$.
- $\omega_i$, angle defined by atoms $C\alpha_i$-$C_i$-$N_{i+1}$-$C\alpha_{i+1}$.

Bond angles:

- $\alpha_i$, bond angle $N_i$-$C\alpha_i$-$C_i$.
- $\beta_i$, bond angle $C_{i-1}$-$N_i$-$C\alpha_i$.
- $\omega_i$, bond angle $C\alpha_i$-$C_i$-$N_{i+1}$.

The distributions of these six angles were computed separately for the AFDB and PDB structure sets. To quantify the divergence between these distributions, we employed several distance and similarity metrics: Kullback-Leibler (KL) divergence, Wasserstein distance (Earth Mover's Distance), Euclidean distance, and Cosine similarity.

Table 6 presents these comparison metrics. The results indicate noticeable differences between the AFDB and PDB angle distributions. For instance, the dihedral angles $\psi$ and $\phi$, which define the Ramachandran plot, exhibit relatively larger KL divergences and Euclidean distances, and lower cosine similarities, compared to the bond angles ($\alpha$, $\beta$, $\gamma$). This suggests that AFDB predictions, while generally accurate, may possess subtly different conformational preferences or a more restricted sampling of these crucial dihedral spaces compared to the ensemble of experimental structures.

Table 6: Comparison of different dihedral angle distributions using various distance metrics.

| | KL ($\times 10^{-7}$) | Wasserstein ($\times 10^{-8}$) | Euclidean Distance | Cosine Similarity |
|---|---|---|---|---|
| $\phi$ | 1.5663 | 5.0828 | 333.0086 | 0.9764 |
| $\psi$ | 3.6680 | 2.5720 | 871.7690 | 0.9349 |
| $\omega$ | 2.6013 | 5.2714 | 355.6580 | 0.9894 |
| $\alpha$ | 0.0057 | 3.5109 | 86.5567 | 0.9950 |
| $\beta$ | 0.0043 | 2.7487 | 75.5303 | 0.9964 |
| $\gamma$ | 0.0012 | 1.8125 | 40.1267 | 0.9991 |

## E.2 Qualitative Analysis

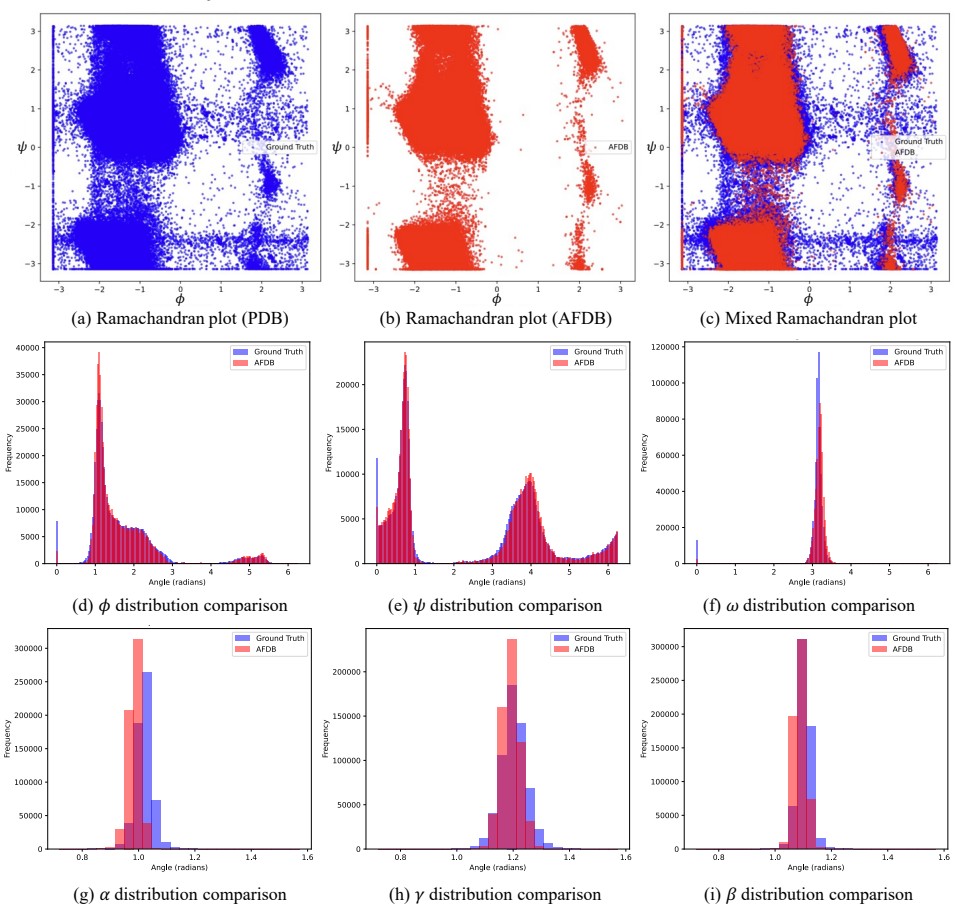

(a) Ramachandran plot (PDB)  (b) Ramachandran plot (AFDB)  (c) Mixed Ramachandran plot

(d) $\phi$ distribution comparison  (e) $\psi$ distribution comparison  (f) $\omega$ distribution comparison

(g) $\alpha$ distribution comparison  (h) $\gamma$ distribution comparison  (i) $\beta$ distribution comparison

Figure 10: Global feature visualization of dihedral angles and bond angles.

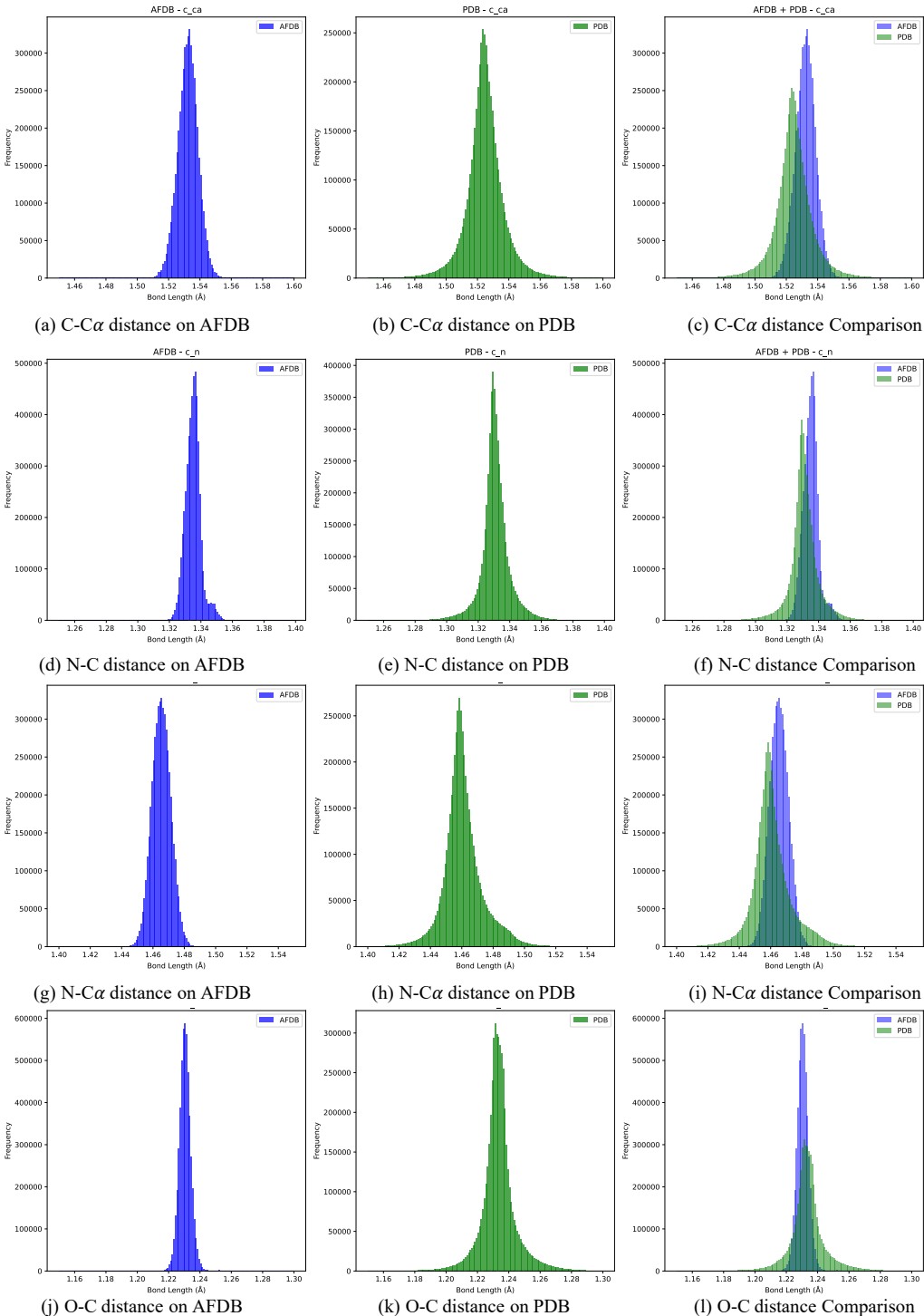

Figure 11: Bond length visualization.

# F  Visualization

To provide a qualitative perspective on the structural modifications introduced by our DeSAE, we visualized representative protein structures from PDB, their corresponding AFDB predictions, and the DSAE-processed Debiased AFDB versions. Figure 12 presents several such examples, with structures superimposed for comparison. At a global level, visual inspection reveals that both the raw AFDB predictions and the Debiased AFDB structures maintain high fidelity to the experimental PDB structures, exhibiting similar overall folds and tertiary arrangements.

These visualizations underscore a key aspect of our findings: **the systematic biases in AFDB that impede inverse folding performance are often not readily apparent through casual visual inspection of global structure**. The differences, while quantitatively significant for downstream deep learning models sensitive to local geometric details, can be quite subtle.

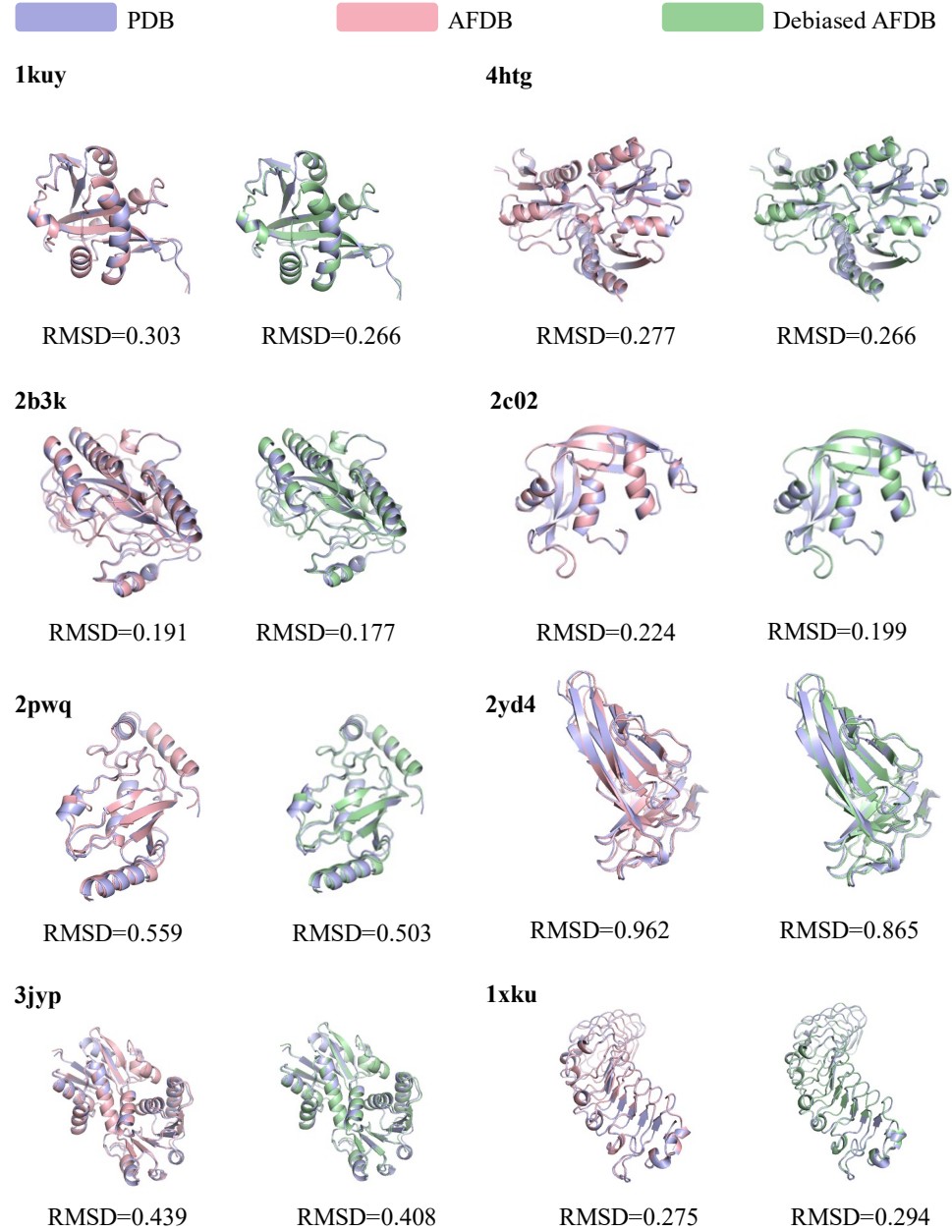

Figure 12: The visualization of samples in PDB, AFDB, and Debiased AFDB.

