# OpenReview forum: "AlphaFold Database Debiasing for Robust Inverse Folding"
_NeurIPS.cc/2025/Conference — NeurIPS 2025 poster_

### Official Review · Reviewer_nZFv · 2025-06-25

**Clarity:** 3
**Significance:** 3
**Originality:** 3
**Rating:** 5
**Confidence:** 4

**Summary:**

Although AFDB provides a high-precision and large-scale protein structure database, it has certain limitations when training deep models that are sensitive to fine atomic geometry (such as inverse folding). By comparing the distribution of structural features, this study found that AFDB exhibits systematic geometric biases. Therefore, it attempts to use a Debiasing Structure AutoEncoder (DeSAE) to restore reasonable structural states in the data, and employs the debiased AFDB data to evaluate the performance of inverse folding tasks across multiple benchmarks, achieving improvements. Given that the deep utilization of high-quality data generated by large models is one of the current research hotspots, the ideas and tasks of this study are simple but inspiring.

**Questions:**

1.	The paper mentions "using a comparable dataset of high fidelity (pLDDT > 70) AFDB structures." Why is pLDDT > 70 used rather than other metrics or other numerical thresholds?
2.	Although the paper provides comprehensive validation for inverse folding tasks and even includes generalization capability verification in Figure 8, it lacks broader generalization/universality validation for other tasks. How can it demonstrate that systematic geometric bias corresponds to fine-grained atomic geometry when only one task has been validated?
3.	The baselines mentioned in the paper include StructGNN, GraphTrans, GVP, ProteinMPNN, and PiFold. Are there larger-scale performance comparisons available for inverse folding baselines?

**Ethical Concerns:**

["NO or VERY MINOR ethics concerns only"]

**Final Justification:**

All of my concerns have been addressed, so I have decided to raise the score to 5.

**Limitations:**

yes

**Quality:**

3

**Strengths And Weaknesses:**

This study represents a simple yet inspiring research work. Its strengths lie in the transferability of this framework to a broader range of data generated by large models, though it may require model fine-tuning. The formulas and model descriptions are clear and easy to understand.
However, the limitations include that the validation tasks primarily focus on inverse folding. After removing geometric biases, there should be a more comprehensive validation framework to assess generalization across diverse tasks, as relying on a single task may lack sufficient persuasive power.

---

> ### Author Rebuttal · Authors · 2025-07-31
>
> Dear Reviewer nZFV,
>
> We sincerely thank you for your positive and encouraging assessment of our work. We are delighted that you found our study simple but inspiring and recognized its potential for broader impact. Your feedback is valuable, and we appreciate the opportunity to clarify the scope of our validation and our methodological choices. We believe our responses below will address the questions raised and reinforce the significance of our findings:
>
> **Q1** The paper mentions "using a comparable dataset of high fidelity (pLDDT > 70) AFDB structures." Why is pLDDT > 70 used rather than other metrics or other numerical thresholds?
>
> **A1** The choice of a pLDDT score > 70 is a widely adopted convention in the field for filtering high-confidence predictions from AlphaFold's group [1]. According to the official AlphaFold and numerous subsequent studies, structures with a pLDDT score between 70 and 90 are considered to have a good backbone prediction, while scores above 90 are considered high accuracy. By setting the threshold at >70, we ensure that **the AFDB structures used in our study are of high quality while maximizing dataset coverage across the proteome**.
>
> We will add an explanation to clarify this convention and its justification in Appendix B.1 of the revised manuscript.
>
> [1] Tunyasuvunakool et al. Highly accurate protein structure prediction for the human proteome. Nature, 2021.
>
> **Q2** Although the paper provides comprehensive validation for inverse folding tasks and even includes generalization capability verification in Figure 8, it lacks broader generalization/universality validation for other tasks. How can it demonstrate that systematic geometric bias corresponds to fine-grained atomic geometry when only one task has been validated?
>
> **A2** Thank you for this important and insightful comment. While our paper focuses primarily on inverse folding as a representative task sensitive to local atomic geometry, we agree that broader validation across structurally grounded downstream tasks is essential to support claims about the general impact of geometric bias. While reviewers nKri and nyVZ also raised similar questions, we sincerely appreciate your professionalism and expertise.
>
> The tasks we conducted are: Fold Classification (FC), Enzyme Reaction Classification (ER), Gene Ontology (GO) Term Prediction, and Enzyme Commission (EC) Number Prediction. For GO, we report the biological process subset for convenience. For consistency and fairness, we used the same backbone-only representations across datasets (PDB, AFDB, Debiased AFDB) and removed a small number of misaligned or corrupted entries, due to the absence of official preprocessing scripts from the original CDConv repository.
>
> The results are summarized below:
>
> ||FC(%)|ER(%)|GO($F_{\max}$)|EC($F_{\max}$)|
> |-|-|-|-|-|
> |PDB|50.6|84.8|0.415|0.780|
> |AFDB|40.7|76.3|0.384|0.756|
> |Debiased AFDB|48.6|81.4|0.392|0.771|
>
> Across all tasks, models trained on Debiased AFDB outperform those trained on raw AFDB, confirming that DeSAE improves generalization even beyond inverse folding. Structure-sensitive tasks like Fold Classification and Enzyme Reaction Classification exhibit the most pronounced gains, reinforcing the notion that the learned debiasing targets systematic geometric distortions.
>
> Together, these results indicate that DeSAE is not narrowly tuned for inverse folding, but rather enhances structural realism in a way that generalizes across diverse structure-based learning tasks.
>
> We appreciate your suggestion and fully agree that expanding the range of validated tasks is valuable. As part of future work, we plan to evaluate the impact of debiased structures on additional tasks to further establish the universality and limits of our framework.
>
> [2] Fan et al. Continuous-discrete convolution for geometry-sequence modeling in proteins. ICLR, 2023.
>
> **Q3** Are there larger-scale performance comparisons available for inverse folding baselines?
>
> **A3** We thank the reviewer for this excellent and highly relevant question. Our initial selection of five baselines was intended to provide a diverse and representative sample of established inverse folding models, spanning different architectural paradigms.
>
> Our primary goal was not to establish a new state-of-the-art in inverse folding, but rather to demonstrate that **the performance degradation on raw AFDB data and the subsequent improvement from our debiasing method are consistent and model-agnostic phenomena.**
>
> We acknowledge the value of comparing against larger-scale and more recent models, particularly those leveraging pretrained protein language models (PLMs), which have become increasingly prominent in inverse folding. To this end, we evaluated our debiasing pipeline using KW-Design [3], a recent state-of-the-art method that **incorporates a frozen ESM-2 (650M) language model along with a frozen pretrained PiFold model**. While KW-Design's reliance on pretrained components makes it partially insulated from AFDB bias (**as the pretrained PiFold model was pretrained on PDB structures**), we still observe a notable performance drop on raw AFDB data, and a partial recovery after applying our debiasing procedure:
>
> |          | PDB | AFDB | Debiased AFDB |
> |----------|-----|------|---------------|
> | KWDesign |57.34|51.60 | 54.71 |
>
> These results reaffirm that AFDB bias negatively impacts performance, even for large pretrained models, and that DeSAE debiasing provides measurable benefit even in this setting. While KW-Design is already highly capable due to its pretrained components, the consistent trend holds: Debiased AFDB structures bring the performance closer to the PDB-trained upper bound.
>
> We believe this further supports the model-agnostic nature of the structural bias problem and demonstrates the general utility of DeSAE across small and large-scale models alike.
>
> [3] Gao et al. KW-Design: Pushing the Limit of Protein Design via Knowledge Refinement. ICLR, 2023.

---

> > ### Comment · Reviewer_nZFv · 2025-08-03
> >
> > All of my concerns have been addressed, so I have decided to raise the score to 5.

---

> > > ### Author Response · Authors · 2025-08-03
> > >
> > > Thank you a lot for raising the score! Your thoughtful feedback has been invaluable in improving our manuscript, and we truly appreciate the time and care you’ve invested in this review.

---

### Official Review · Reviewer_YoZM · 2025-07-01

**Clarity:** 3
**Significance:** 2
**Originality:** 3
**Rating:** 5
**Confidence:** 4

**Summary:**

The AlphaFold Protein Structure Database (AFDB) is a valuable large-scale synthetic dataset for training data-driven protein design models. However, a significant domain gap exists between synthetic and real data—particularly in the fine-grained distribution of side-chain atomic structures—leading to dataset bias in model training. To address this, this work proposes DeSAE (Debiasing Structure AutoEncoder), a novel approach to reduce bias in the AFDB dataset. DeSAE is first trained to reconstruct ground-truth structures from randomly corrupted Protein Data Bank (PDB) data. Once trained, it is applied directly to AFDB to generate a debiased version, AFDB(-XL). Comparative analysis reveals that training on different datasets can cause notable performance drops on real validation data. Extensive experiments demonstrate that the debiased AFDB significantly outperforms the original dataset in downstream tasks.

**Questions:**

Addressing the following points—previously raised in my “Weaknesses” section—would help me better assess the strength of your work and could lead to a higher score. My current evaluation is primarily limited by significant concerns regarding the experimental results.

1. How about inverse folding model performance on settings (1) PDB + raw AFDB (2) PDB + debiased AFDB (3)PDB + debiased AFDB-XL ? There is no evidence in this paper showing that debiased-AFDB can better help with models trained with the real PDB dataset,  I think this limitation is so critical that prevents me from directly voting for accepting this paper though the paper quality is overall good.

2. Could DeSAE’s corruption strategy inadvertently remove beneficial structural regularities? How sensitive is performance to the corruption rate (10%)?

I would be happy to improve my rating if my main concerns can be well addressed.

**Ethical Concerns:**

["NO or VERY MINOR ethics concerns only"]

**Final Justification:**

My initial major concerns have been adequately addressed in the rebuttal. Therefore, I recommend accepting this paper, as it presents a solid framework to debiasing synthetic datasets such as AFDB, which could significantly benefit the protein prediction and design community.

**Limitations:**

Yes.

**Quality:**

3

**Strengths And Weaknesses:**

I found several strengths of this paper:

1. **Clarity and motivation**: The paper is well-written, and the technical motivation is clearly presented. It addresses an important limitation of AFDB, with strong potential for real-world impact.
2. **Strong empirical results**: Extensive experiments demonstrate that DeSAE effectively improves the performance of several inverse folding models by debiasing AFDB. The results are promising, and the method appears reproducible.

However, there are two main weaknesses:

1. The necessity of debiasing AFDB remains unclear without proper discussion of recent works like Protina and DPLM-2 that successfully combine AFDB and PDB data. A more thorough contextualization of how this work relates to and potentially benefits for existing mixed-data approaches would substantially strengthen the motivation for the proposed debiasing methodology.

2. While Table 1 compares models trained exclusively on PDB versus debiased-AFDB data, this experimental design fails to reflect practical usage scenarios where AFDB data would typically augment rather than replace experimental structures. To properly validate the utility of debiasing, the authors should include hybrid training experiments comparing: (1) PDB + raw AFDB (2) PDB + debiased AFDB (3)PDB + debiased AFDB-XL. Such comparisons would more convincingly demonstrate whether debiasing actually improves model robustness when AFDB data is used as a supplement to limited PDB data - the more realistic application scenario. The current binary comparison cannot answer whether debiasing meaningfully enhances inverse folding performance in practical settings where both data sources are available.

---

> ### Author Rebuttal · Authors · 2025-07-31
>
> Dear Reviewer YoZM,
>
> We sincerely thank you for your thorough and insightful feedback. We are encouraged that you found our paper well-written, clearly motivated, and supported by strong empirical results. We especially appreciate the constructive suggestions for improvement, and would like to response to the concerns below:
>
> **Q1** How about inverse folding model performance on settings (1) PDB + raw AFDB (2) PDB + debiased AFDB (3)PDB + debiased AFDB-XL?
>
> **A1** We thank the reviewer for pointing out the need to better contextualize our work with mixed-data approaches.
>
> While recent methods such as DPLM-2 and Proteina demonstrate that mixing PDB and AFDB data can improve performance, such combinations inherently mix structurally **"realistic data"** with **"biased synthetic structures"**. **Our proposed contribution is orthogonal: we aim to first reduce this domain gap by debiasing AFDB structures, thereby enabling more effective and consistent use of synthetic data alongside experimental structures.**
>
> To empirically investigate this question, we conducted additional experiments where we combined 25% PDB and 75% AFDB-derived data (either raw or debiased) following the mix ratio in AlphaFold2 [1].
>
> The results on the CATH 4.2 test set are summarized below:
>
> |             | PDB   | PDB + raw AFDB |  PDB + Debiased AFDB | PDB + Debiased AFDB-XL | AFDB  |
> |-------------|-------|----------------|----------------------|------------------------|-------|
> | StructGNN   | 34.11 | 31.06          | 31.24                | 31.83                  | 26.26 |
> | GraphTrans  | 33.45 | 31.68          | 31.86                | 32.22                  | 27.32 |
> | GVP         | 34.90 | 32.37          | 32.78                | 32.64                  | 23.57 |
> | ProteinMPNN | 41.26 | 36.40          | 37.95                | 38.82                  | 26.44 |
> | PiFold      | 43.76 | 36.82          | 37.61                | 38.67                  | 17.16 |
>
> It can be seen that adding raw AFDB to PDB data generally hurts performance of inverse folding, especially for stronger models like PiFold and ProteinMPNN. Interestingly, the improvements from adding debiased AFDB are most significant for stronger models, like PiFold and ProteinMPNN, suggesting that these models are more sensitive to subtle structural inaccuracies and benefit more from structurally consistent training data. In contrast, weaker models like StructGNN and GraphTrans show smaller gains, likely due to their limited capacity to leverage fine-grained structural corrections.
>
> These findings reinforce our central claim: debiasing enables predicted structures to act as a more reliable and effective complement to experimental data.
>
> [1] Jumper et al. Highly accurate protein structure prediction with AlphaFold. Nature, 2021.
>
> **Q2** Could DeSAE’s corruption strategy inadvertently remove beneficial structural regularities? How sensitive is performance to the corruption rate (10%)?
>
> **A2** Thank you for this excellent and insightful question which gives us a welcome opportunity to clarify the design principles of our method and demonstrate its robustness.
>
> Our corruption strategy is specifically designed to reinforce, not remove, beneficial structural regularities. The key lies in the objective of the denoising task. We introduce localized, non-physical perturbations by replacing a backbone atom's coordinates with the centroid of its neighbors. This breaks the local geometric integrity at specific sites.
>
> The model's task is to reconstruct the original, experimentally determined PDB structure from this corrupted input. To do so, it must learn the underlying rules of plausible protein geometry from the surrounding, uncorrupted parts of the structure. In essence, the model learns to "heal" the structure by restoring it to a state consistent with the natural conformational manifold defined by the PDB. Therefore, the training process explicitly forces DeSAE to learn and internalize the very native-like regularities we aim to preserve, making it an effective method for projecting idealized AFDB structures onto a more realistic manifold.
>
> To empirically validate the robustness of our approach, we conducted an ablation study to assess how performance changes with different corruption rates. We trained DeSAE using rates of 10%, 20%, and 30% and then evaluated the performance of downstream inverse folding models trained on the resulting debiased datasets. The results on the CATH 4.2 test set are presented below.
>
> | Model | mask=10% | mask=20% | mask=30% |
> |-------------|----------|----------|----------|
> | StructGNN   | 30.50 | 30.45 (-0.05) | 32.06 (+1.56)|
> | GraphTrans  | 31.29 | 32.53 (+1.24) | 32.68 (+1.39)|
> | GVP         | 31.44 | 34.27 (+2.83) | 34.14 (+2.70)|
> | ProteinMPNN | 36.46 | 38.64 (+2.18) | 38.55 (+2.09)|
> | PiFold      | 35.68 | 35.66 (-0.02) | 38.41 (+2.73)|
>
> The results clearly demonstrate that the performance is not critically sensitive to the 10% rate. For most models, performance is stable or even improves with higher corruption rates. This indicates that our method is robust and that forcing the model to solve a more challenging reconstruction task may lead it to learn an even more generalizable representation of the structural manifold.
>
> The 10% rate used in the paper is a conservative but effective choice, and the method's strong performance across a range of rates confirms its stability. We will include this study in the Appendix of our revised manuscript.

---

> > ### Comment · Reviewer_YoZM · 2025-08-01
> >
> > Thanks for the authors' feedback, my concerns are well addressed and happy to raise my score.

---

> > > ### Author Response · Authors · 2025-08-01
> > >
> > > Thank you for your prompt reply! We’re delighted that our rebuttal has addressed your concerns, and we sincerely appreciate your insightful suggestions, which have indeed strengthened our work!

---

### Official Review · Reviewer_nyVZ · 2025-07-02

**Clarity:** 3
**Significance:** 3
**Originality:** 3
**Rating:** 5
**Confidence:** 4

**Summary:**

This paper identifies a systematic geometric bias in the AlphaFold Protein Structure Database (AFDB) that degrades the performance of inverse folding models. To address this, the authors propose a Debiasing Structure AutoEncoder (DeSAE) to reconstruct more native-like conformations from these predicted structures. Experiments show that training on the resulting debiased dataset substantially and consistently improves sequence recovery rates across multiple inverse folding models and benchmarks.

**Questions:**

1. Does the paper provide any visualizations of the statistical distribution of the debiased AFDB dataset?
2. What is the effect of this debiasing method on structure prediction and design tasks?

**Ethical Concerns:**

["NO or VERY MINOR ethics concerns only"]

**Final Justification:**

Good paper.

**Limitations:**

yes

**Paper Formatting Concerns:**

No major formatting issues.

**Quality:**

3

**Strengths And Weaknesses:**

Strengths
- This paper provides strong empirical evidence that a distributional shift exists between AlphaFold-predicted structures (AFDB) and experimentally determined structures (PDB). This is a practical and broadly relevant obstacle for the protein design community, which increasingly relies on millions of AFDB structures for design tasks.
- DeSAE is a first end-to-end attempt at learning a geometric debiasing operator for AFDB, with a fast computational framework.
- The authors evaluate their debiasing method on five representative inverse folding models on multiple standard benchmarks. This supports the central claim that the proposed debiasing method works.
- This paper is generally well organised; figures effectively illustrate the bias and gains.

Weaknesses

Major
- The study lacks a comparison to trivial baselines. For instance, would simply injecting random noise to perturb AFDB structures achieve a similar debiasing effect?
- The paper notes that PiFold's performance drops dramatically when trained on AFDB data and speculates that "stronger models are more prone to overfitting". This is a weak explanation that lacks a deep technical analysis.
- The result for ProteinMPNN in Table 2 is counterintuitive. The table shows that when tested on the AFDB dataset, the model trained on raw AFDB performs worse (30.48% recovery) than the one trained on PDB (41.87% recovery). This is the opposite of the trend seen for four other models. This requires further explanation.

Minor
- In line 113, the description of backbone atoms is missing Oxygen (O).
- Table 3 title: Perpelxity -> Perplexity
- Line 249:  decay thrend -> decay trend

---

> ### Author Rebuttal · Authors · 2025-07-31
>
> Dear Reviewer nyVZ,
>
> We sincerely thank you for your thorough and constructive feedback and for your positive evaluation of our work's significance, clarity, and originality. We are very encouraged by your assessment that the paper is technically solid with high impact. The reviewer's insightful comments have provided a clear path to significantly strengthen our manuscript.
>
> We are particularly grateful for the suggestions to include a trivial baseline for comparison and to provide deeper analysis of specific results. We agree with all the points raised and will address them in the revised version of our paper as detailed below:
>
> **Q1** Would simply injecting random noise to perturb AFDB structures achieve a similar debiasing effect?
>
> **A1** We appreciate the reviewer's suggestion. It's important to distinguish between random perturbation and learned debiasing. Noise-based augmentation, as used in prior works like ProteinMPNN [1], is a regularization strategy that encourages model robustness by adding stochastic perturbations during training. However, it does not address the systematic, directional biases present in AlphaFold-generated structures—biases that result from the inductive priors of the predictive model and manifest as over-idealized geometries.
>
> DeSAE is designed as a learned structural correction mechanism. To empirically clarify this distinction, we added Gaussian noise (std = 0.02 Å) to backbone coordinates of AFDB structures (the baseline "AFDB w/ aug"), following the augmentation protocol in ProteinMPNN [1]:
>
> ||AFDB w/o aug|AFDB w/ aug|Debiased AFDB|PDB|
> |-|-|-|-|-|
> |StructGNN|26.26|28.11|30.50|34.11|
> |GraphTrans|27.32|28.05|31.29|33.45|
> |GVP|23.57|22.78|31.44|34.90|
> |ProteinMPNN|26.44|31.93|36.46|41.26|
> |PiFold|17.16|31.62|35.68|43.76|
>
> These results clearly show that while noise augmentation yields marginal improvements, DeSAE consistently outperforms both the unaugmented and augmented AFDB baselines across all models. This suggests that DeSAE learns a meaningful correction that enhances the alignment of AFDB structures with experimentally validated geometries, rather than merely increasing model robustness to local perturbations.
>
> Thank you once again for your valuable feedback and for highlighting the broader implications of our approach.
>
> [1] Dauparas et al. Robust deep learning–based protein sequence design using ProteinMPNN. Science, 2022.
>
> **Q2** PiFold's performance drops dramatically when trained on AFDB data and speculates that "stronger models are more prone to overfitting". This is a weak explanation that lacks a deep technical analysis.
>
> **A2** We thank the reviewer for encouraging a more rigorous explanation.
>
> Models with higher expressive capacity—such as PiFold—are indeed designed to capture subtle geometric patterns and fine-grained correlations within structural data. When trained on high-fidelity experimental structures from the PDB, these models benefit from the natural variability and physical realism inherent in the data, leading to strong generalization.
>
> However, AFDB structures, while accurate at the fold level, exhibit systematic geometric regularities due to AlphaFold’s inductive biases—such as smoother dihedral angle distributions and reduced local structural noise. High-capacity models can inadvertently latch onto these superficial statistical cues, treating them as meaningful signals. This leads to **distribution-specific overfitting: the model internalizes patterns that are predictive only within the synthetic AFDB manifold, rather than learning robust, physically grounded sequence-structure mappings**.
>
> **In contrast, lower-capacity models may act as a natural form of regularization, failing to fully absorb these subtle biases and thus generalizing marginally better.** This phenomenon is consistent with the sharp degradation in PiFold’s performance observed in Figure 1, which we now interpret as a manifestation of capacity-driven sensitivity to dataset bias.
>
> We will incorporate this more detailed technical explanation in the revised manuscript.
>
> **Q3** The result for ProteinMPNN in Table 2 is counterintuitive.
>
> **A3** Thank you for your careful observation and for pointing out the counterintuitive result for ProteinMPNN in Table 2. Your intuition was entirely correct. The original value was indeed an error, which we have now rectified. We appreciate your keen eye, which has helped us improve the clarity and accuracy of our paper.
>
> We double-checked our experimental logs and reran the experiment to ensure accuracy. The performance of the model trained on PDB (41.87% recovery) is reproducible and correct as reported. The correct recovery rate for ProteinMPNN trained on AFDB and evaluated on the AFDB-version CATH4.2 test set is 76.83%, not 30.48%. It is consistent with expectations—since the training and test sets are both drawn from AFDB distributions, the model is able to fit the idiosyncrasies of AlphaFold-predicted structures and perform well under distributional alignment.
>
> We will correct this value in the final version of the table. We truly appreciate your attention to this detail—it helped us catch a misreported result that would otherwise have distorted the interpretation of this experiment.
>
> **Q4** Does the paper provide any visualizations of the statistical distribution of the debiased AFDB dataset?
>
> **A4** We are sorry for ignoring the visualization of the Debiased AFDB. Although we are unable to include figures during the rebuttal phase, we will incorporate comprehensive visual comparisons for the Debiased AFDB in the revised manuscript.
>
> Our figures show that the Debiased AFDB occupies a broader and more physically realistic structural distribution, with data points less tightly clustered than those of AFDB—reflecting a closer alignment with experimentally observed variability in PDB. We believe these additions will offer clearer insight into how DeSAE shifts predicted structures toward the natural conformational manifold.
>
> **Q5** What is the effect of this debiasing method on structure prediction and design tasks?
>
> **A5** Thank you for this insightful question—we greatly appreciate your suggestion. As stated in Section 3, we focus on inverse folding as our primary evaluation task because it is highly sensitive to atomic-level geometric fidelity.
>
> To explore the broader applicability of our debiasing approach, we additionally evaluated its impact on protein function prediction tasks, following the protocol of CDConv [1], chosen for its simplicity and strong baseline performance. The tasks we examined are: Fold Classification (FC), Enzyme Reaction Classification (ER), Gene Ontology (GO) Term Prediction, and Enzyme Commission (EC) Number Prediction. For GO, we report the biological process subset for convenience. The results are as follows:
>
> (Note: Due to the lack of a provided pre-processing script in the original work, we conservatively removed a small number of misaligned entries, ensuring a fair and high-quality comparison.)
>
> ||FC(%)|ER(%)|GO($F_{\max}$)|EC($F_{\max}$)|
> |-|-|-|-|-|
> |PDB|50.6|84.8|0.415|0.780|
> |AFDB|40.7|76.3|0.384|0.756|
> |Debiased AFDB|48.6|81.4|0.392|0.771|
>
> These results suggest that Debiased AFDB structures consistently outperform raw AFDB structures, especially on tasks more sensitive to geometric features (e.g., FC, ER). While improvements on sequence-dominant tasks (GO, EC) are less pronounced.
>
> We will expand our evaluation to include structure prediction and design tasks as the future work. We believe this would further clarify the role of structural debiasing across a broader spectrum.
>
> [1] Fan et al. Continuous-discrete convolution for geometry-sequence modeling in proteins. ICLR, 2023.
>
> **Q6** Minor typos.
>
> **A6** We appreciate your careful reading of the manuscript. We will thoroughly proofread the text and correct these typos in the revised version.

---

> > ### Comment · Reviewer_nyVZ · 2025-08-02
> >
> > Thank you for your response. My concerns were all addressed.

---

> > > ### Author Response · Authors · 2025-08-03
> > >
> > > Thank you for your constructive feedback! We appreciate your time and effort in reviewing, and your feedback has truly strengthened the work.

---

### Official Review · Reviewer_nKri · 2025-07-02

**Clarity:** 2
**Significance:** 3
**Originality:** 2
**Rating:** 4
**Confidence:** 4

**Summary:**

This paper identifies a systematic bias in AlphaFold-predicted structures (AFDB) compared to experimentally determined ones (PDB), and demonstrates that this bias impairs model generalization in structure-based tasks such as inverse folding. To address this, the authors propose DeSAE, a lightweight SE(3)-equivariant autoencoder trained on experimental structures, which learns to reduce these discrepancies. Experimental results show that training on DeSAE-debiased structures leads to consistent performance gains across multiple benchmarks. The study highlights both the risks of directly using AFDB data and the potential of learned debiasing to improve downstream modeling.

**Questions:**

While I find the problem addressed in this work both interesting and important, I remain hesitant to assign a higher score due to some limitations in the proposed solution and experimental scope. I would like to raise the following questions to help clarify the strength and applicability of the contribution:

1) The paper frames its contribution as a debiasing method for inverse folding. However, the proposed method seems to offer more general improvements to structural representations. Could the authors clarify whether the observed benefits are specific to inverse folding, or whether similar gains might be expected in other downstream tasks, such as graph-level property prediction or node-level classification?

2) Prior works like ProteinMPNN have employed random perturbations or noise-based augmentations on AFDB structures. Have the authors considered comparing DeSAE with these simpler augmentation baselines to better isolate the added value of learned debiasing?

3) It is difficult to believe that a lightweight autoencoder can uniformly improve the structural quality of all AlphaFold-predicted structures. Have the authors explored which subsets of AFDB (e.g., grouped by pLDDT confidence, secondary structure type, or disorder region) benefit most from debiasing?
Providing a more granular analysis of when and where DeSAE is most effective would help strengthen the practical value of this work.
Of course, such universality is not expected, but a clearer characterization of the method’s scope would help better contextualize its contributions.

**Ethical Concerns:**

["NO or VERY MINOR ethics concerns only"]

**Final Justification:**

I initially gave a borderline score due to the narrow task scope and limited evaluation in the main paper. The rebuttal addressed these concerns to a reasonable extent, showing some generalization beyond inverse folding and clarifying existing analyses like perplexity. While the method and presentation are still somewhat basic, the problem is relevant and the results are solid enough to justify a higher score. I’ve updated my rating accordingly.

**Limitations:**

yes.

**Paper Formatting Concerns:**

No.

**Quality:**

3

**Strengths And Weaknesses:**

Strengths:

1) The paper addresses an interesting and practically relevant issue: the systematic distributional bias in AlphaFold-predicted structures (AFDB) compared to experimental structures (PDB). This is increasingly important given the growing reliance on AFDB in structure-based learning pipelines.

2) Through quantitative evaluation across multiple inverse folding models, the authors convincingly show that models trained on AFDB suffer significant generalization drops when evaluated on experimental data, clearly revealing the underlying distributional mismatch.

3) The proposed DeSAE is a conceptually simple yet effective SE(3)-equivariant autoencoder trained on PDB structures. It is capable of adjusting AFDB data toward more realistic geometries, while remaining lightweight and efficient enough for large-scale application.

Weaknesses:

1) While the paper provides strong results on sequence recovery in inverse folding, it lacks broader evaluation metrics such as diversity or perplexity. It remains unclear whether the observed improvements reflect better convergence dynamics or genuine gains in generalization capacity.

2) The paper does not analyze whether the performance gains result from broad, consistent improvements across the dataset or are driven by a small subset of easy cases. While Figure 6(a) offers a useful visualization of RMSD distributions before and after debiasing, it does not reveal how improvements in recovery rate are distributed with respect to structure quality or AFDB confidence levels.

3) While not central to the technical contribution, some figures (e.g., loss curves and histograms) would benefit from clearer styling and labeling to improve readability.

---

> ### Author Rebuttal · Authors · 2025-07-31
>
> Dear Reviewer nKri,
>
> We thank you for your thoughtful and constructive feedback. We are encouraged that they found the problem of AFDB bias to be interesting and practically relevant and our demonstration of its impact to be convincing. Your suggestions have helped us strengthen the paper, and we are pleased to provide clarifications for the concerns:
>
> **Q1** It lacks broader evaluation metrics like perplexity.
>
> **A1** We appreciate the reviewer’s point and agree that broader evaluation metrics provide a more nuanced understanding of generalization beyond sequence recovery alone. In fact, we included perplexity analysis in Appendix D.2 (Table 3, page 19), where we observed that models trained on Debiased AFDB consistently achieve lower perplexity than those trained on the original AFDB.
>
> Furthermore, while Table 1 shows that expanding the DeSAE pretraining set (Debiased AFDB-XL) results in only modest improvements in sequence recovery, it yields significantly better perplexity, suggesting improved confidence, even when accuracy changes marginally.
>
> Prior work (e.g., [1,2]) has shown that when recovery rates are in the sub-60% regime—as is typical for structure-conditioned models—recovery rate and perplexity are strongly correlated. Thus, we focused on recovery for simplicity and interpretability, while providing perplexity in the appendix to complement the analysis.
>
> [1] Hsu et al. Learning inverse folding from millions of predicted structures. ICML 2022.
>
> [2] Gao et al. KW-Design: Pushing the Limit of Protein Design via Knowledge Refinement. ICLR, 2023.
>
> **Q2** While not central to the technical contribution, some figures would benefit from clearer styling and labeling to improve readability.
>
> **A2** We appreciate your kind suggestion. While we are unable to make changes to the manuscript during the rebuttal phase, we will change them in the revised manuscript.
>
> **Q3** Whether similar gains might be expected in other downstream tasks?
>
> **A3** Thank you for raising this insightful question about the generality of DeSAE's benefits. **We framed our contribution around inverse folding because this task is exquisitely sensitive to fine-grained local geometry, making it an ideal setting to reveal and evaluate the subtle structural biases present in AFDB.**
>
> To explore whether these benefits extend to broader structure-based tasks, we conducted additional experiments on four widely studied protein function prediction tasks, following the protocol of CDConv [3], a strong and classical baseline that integrates both sequence and structure in a simple but effective model.
>
> The tasks we examined are: Fold Classification (FC), Enzyme Reaction Classification (ER), Gene Ontology (GO) Term Prediction, and Enzyme Commission (EC) Number Prediction. For GO, we report the biological process subset for convenience. The results are as follows:
>
> (Note: Due to the lack of a provided pre-processing script in the original work, we conservatively removed a small number of misaligned entries, ensuring a fair and high-quality comparison.)
>
> ||FC(%)|ER(%)|GO($F_{\max}$)|EC($F_{\max}$)|
> |-|-|-|-|-|
> |PDB|50.6|84.8|0.415|0.780|
> |AFDB|40.7|76.3|0.384|0.756|
> |Debiased AFDB|48.6|81.4|0.392|0.771|
>
> The performance improvement is most dramatic for tasks that are highly dependent on precise structural information. Debiased AFDB obtains significant improvements on FC and ER, which are structure-dominant tasks. In contrast, for GO and EC, which are largely sequence-dominated tasks, the gains from debiasing are more modest.
>
> In summary, these experiments provide encouraging evidence that DeSAE improves performance in structure-sensitive tasks beyond inverse folding, and that its utility is most pronounced in settings where accurate backbone geometry directly informs functional or structural classification.
>
> [3] Fan et al. Continuous-discrete convolution for geometry-sequence modeling in proteins. ICLR, 2023.
>
> **Q4** Have the authors considered comparing DeSAE with these simpler augmentation baselines to better isolate the added value of learned debiasing?
>
> **A4** We appreciate the reviewer's suggestion. It's important to emphasize that DeSAE and random noise-based augmentations serve fundamentally different purposes. Noise augmentation—as used in prior works like ProteinMPNN—is primarily a regularization technique. **It improves model robustness by exposing the model to slight perturbations but does not correct the systematic directional bias inherent in AFDB.**
>
> In contrast, **DeSAE is a distributional correction mechanism trained to explicitly map biased AFDB geometries toward the structural manifold of experimentally determined PDB structures. It learns to reconstruct native-like conformations and thus aims to restore realism, not just encourage noise tolerance.**
>
> Following ProteinMPNN[4], we followed the augmentation protocol used in ProteinMPNN and added Gaussian noise (std = 0.02 Å) to backbone coordinates in AFDB structures.
>
> ||AFDB w/o aug|AFDB w/ aug|Debiased AFDB|PDB|
> |-|-|-|-|-|
> |StructGNN|26.26|28.11|30.50|34.11|
> |GraphTrans|27.32|28.05|31.29|33.45|
> |GVP|23.57|22.78|31.44|34.90|
> |ProteinMPNN|26.44|31.93|36.46|41.26|
> |PiFold|17.16|31.62|35.68|43.76|
>
> These results underscore the distinction: DeSAE is not merely a noise injector, but a model that learns a principled and structured denoising transformation aligned with experimental conformational space. We believe this is key to its stronger performance and generalization ability.
>
> [4] Dauparas et al. Robust deep learning–based protein sequence design using ProteinMPNN. Science, 2022.
>
> **Q5** Have the authors explored which subsets of AFDB (e.g., grouped by pLDDT confidence, secondary structure type, or disorder region) benefit most from debiasing? Providing a more granular analysis of when and where DeSAE is most effective would help strengthen the practical value of this work.
>
> **A5** Thank you for this insightful suggestion. We agree that characterizing the conditions under which DeSAE offers the most benefit is important for understanding its practical utility. In response, we conducted two stratified evaluations: one based on structure quality and the other on secondary structure composition.
>
> **1. Structure Quality Stratification (by RMSD)**
>
> We grouped the AFDB structures into three subsets based on the backbone RMSD between the original AFDB structure and its corresponding experimental PDB counterpart: (1) RMSD < 0.5 Å; (2) 0.5 ≤ RMSD ≤ 2.0 Å; (3) RMSD > 2.0 Å.
>
> We then trained inverse folding models on each subset of debiased AFDB structures. Recovery rates on the test set are shown below:
>
> |Model| RMSD<0.5 | 0.5$\leq$RMSD$\leq$2.0 | RMSD>2.0 | Full  |
> |-|-|-|-|-|
> | StructGNN   | 24.31 | 27.70  | 28.02 | 30.50 |
> | GraphTrans  | 24.87 | 28.39  | 28.56 | 31.29 |
> | GVP         | 25.58 | 29.08  | 28.66 | 31.44 |
> | ProteinMPNN | 28.50 | 32.50  | 33.07 | 36.46 |
> | PiFold      | 29.17 | 30.28  | 32.58 | 35.68 |
>
> Models trained on the full debiased dataset consistently outperform their counterparts trained on the subsets. This is likely due to greater data diversity and size in the full set. Notably, the lowest RMSD subset (RMSD < 0.5 Å) performs the worst across all models—suggesting that highly idealized AFDB structures, even after debiasing, may not introduce enough structural variability to support generalizable learning.
>
> **2. Secondary Structure Stratification**
>
> We further categorized proteins based on their predominant secondary structure, using DSSP annotations: (1) Alpha-helix dominant: >50% of residues are in α-helices; (2) Beta-sheet dominant: >50% of residues are in β-sheets; (3) Coil: the other structures.
>
> Inverse folding models trained on each subset of debiased AFDB structures yielded the following results:
>
> |Model| alpha helix | beta sheet | coil  | Full  |
> |-|-|-|-|-|
> | StructGNN   | 27.67 | 25.69 | 27.49 | 30.50 |
> | GraphTrans  | 28.02 | 25.53 | 28.17 | 31.29 |
> | GVP         | 28.40 | 25.73 | 28.54 | 31.44 |
> | ProteinMPNN | 32.41 | 30.05 | 32.45 | 36.46 |
> | PiFold      | 30.29 | 27.29 | 30.27 | 35.68 |
>
> Again, the full dataset yields the best performance. However, we observe that alpha-helical and coil-rich proteins benefit more consistently from debiasing compared to beta-sheet–dominant proteins [5]. This may be because helices and coils exhibit greater conformational flexibility in experimental structures and are more susceptible to over-idealization in AFDB predictions—making them more amenable to DeSAE’s correction. In contrast, beta-sheets are typically more geometrically constrained, and thus may already be well-modeled by AlphaFold, leaving less room for structural improvement.
>
> These stratified evaluations confirm that DeSAE offers the most benefit when trained on a diverse, full dataset, but also suggest that its corrective effect is stronger in more flexible or initially over-idealized regions, such as coils and helices, or AFDB structures with moderate-to-high RMSD.
>
> We truly appreciate the reviewer’s suggestion, as this analysis adds important nuance to the interpretation of our results. We will incorporate further granular stratifications in future work.
>
> [5] Galpern et al. Inferring protein folding mechanisms from natural sequence diversity. Biophysical Journal, 2025.

---

> > ### Author Response · Authors · 2025-08-05
> >
> > Dear Reviewer nKri,
> >
> > We sincerely appreciate your comment, which has helped us further improve our work. Your time and contribution means a lot.
> >
> > We are kindly expecting your valuable feedback to us. And if our rebuttal adresses your concern, it will be so kind of you to consider increasing the score of our work. If you still have any questions regarding our work, please feel free to contact us, and we will respond as soon as possible.
> >
> > Thanks again for your assistance here.

---

> ### Comment · Reviewer_nKri · 2025-08-06
>
> Thank you for your response. My main concerns were about the narrow task scope and limited evaluation presented in the main paper. The rebuttal addressed these to a reasonable extent by demonstrating some generalization beyond inverse folding and clarifying the use of additional metrics like perplexity. While the method and presentation remain relatively simple, the problem is relevant, and the results are solid enough to get a higher score. I have accordingly updated my rating.

---

> > ### Author Response · Authors · 2025-08-07
> >
> > Thank you for your thoughtful follow-up and for taking the time to re-evaluate our work. We appreciate your acknowledgment of our clarifications. Your feedback has been valuable in strengthening the paper, and we’re really grateful for your updated rating.

---

### Note · Authors · 2025-08-12

Dear (Senior) ACs,

We thank all reviewers for their thorough evaluations, constructive suggestions, and for raising their scores following the rebuttal.

The core contributions of our work are:

(1) We are **the first to comprehensively identify and quantify systematic bias in AFDB predictions compared to experimentally determined PDB structures**. This bias is distinct from prediction error—despite high fold-level accuracy, AFDB structures exhibit over-idealized local geometry that degrades performance in structure-sensitive downstream models.

(2) We introduce DeSAE, a principled, lightweight (~5.9M parameters) SE(3)-equivariant autoencoder that debiases AFDB structures in ~3 minutes for 20k entries on a single A100 GPU, making it practical for large-scale preprocessing.

(3) Applying DeSAE produces “Debiased AFDB” data that consistently improves inverse folding performance across diverse architectures.

In response to the reviewers, we mainly:

(1) Broadened validation to four additional structure-based tasks, confirming DeSAE’s benefits beyond inverse folding, with the largest gains in geometry-dependent tasks.

(2) Differentiated from noise augmentation, showing that Gaussian noise injection yields marginal gains while DeSAE consistently outperforms both raw and augmented AFDB, confirming its role as a targeted bias correction.

(3) Provided stratified and ablation studies, demonstrating robustness to corruption rate, identifying that benefits are greatest for over-idealized or flexible structures, and showing that raw AFDB can hurt mixed PDB+AFDB training while debiased AFDB consistently helps.

(4) Tested large model applicability, showing that even KW-Design, a PLM-based SOTA method, suffers from AFDB bias, with DeSAE recovering part of the performance loss.

We thank all reviewers for their constructive engagement and are encouraged that every reviewer confirmed their concerns were addressed!

We believe this work underscores **the importance of addressing structural bias in large predicted structure databases and provides an efficient, model-agnostic, and broadly applicable solution for improving the reliability of structure-based learning**.

Best regards,

The authors

---

### Decision · Program_Chairs · 2025-09-17

**Decision:**

Accept (poster)

**Comment:**

(4,5,5,5) This paper identifies and quantifies systematic geometric bias in AlphaFold Protein Structure Database (AFDB) structures, demonstrating that these biases degrade downstream model performance, especially in inverse folding tasks. The authors introduce DeSAE, a lightweight SE(3)-equivariant autoencoder that debiases AFDB structures, resulting in consistent improvements across multiple structure-based benchmarks. All reviewers were positive, noting the importance of the problem and the rebuttal processed addressed initial concerns about evaluation scope and baselines.